# Machine learning-based criminal behavior analysis for enhanced digital forensics

**W. Pawani Dananjana**◉, **Jithmi Sewwandi Arambawela**◉,
**D. G. Samesha Navodi Gonawala**◉, **R. K. Gethmi Hasinika Rathnayaka**◉,
**Amila Nuwan Senarathne, S. M. Deemantha Nayanajith Siriwardena**◉*

Faculty of Computing, Sri Lanka Institute of Information Technology, Malabe, Sri Lanka

◉ These authors contributed equally to this work.
* deemantha.s@sliit.lk

## Abstract

In an increasingly digital world, uncovering criminal activity often relies on analyzing vast amounts of online behavior. Traditional methods in digital forensics struggle to keep up with the complexity and volume of data, particularly when trying to detect subtle deviations in online activity that could signal illegal intent. This research introduces an innovative approach that leverages machine learning to analyze internet activity—specifically browser artifacts—shedding new light on criminal behaviors that would otherwise remain hidden.Using advanced machine learning techniques such as Long Short-Term Memory (LSTM) networks and Autoencoders, this study focuses on detecting suspicious patterns and anomalies in browsing activity. By understanding the sequence and timing of a user's online actions, this method enhances digital forensics investigations, allowing for faster and more accurate detection of criminal intent and behavior. The research aims to improve the speed and accuracy of identifying malicious online activity but also offers law enforcement and investigators a powerful tool to make sense of complex data. These findings represent an important step towards advancing digital forensics, enabling deeper insights into criminal behavior and more effective investigations, ultimately contributing to a safer digital environment.

## Introduction

As a result of rapid digitalization, the number of cybercrimes and computer-aided conventional crimes has grown significantly. Consequently, the demand for adaptive digital forensic techniques and tools has also increased [1]. However, in the age of big data, where the volume of digital evidence is substantial, an accurate and time-efficient analysis has become nearly impossible to perform manually.

Criminal behavior analysis has been an invaluable tool for decades in traditional criminal investigations. It has been recognized for its role in reconstructing perpetrators' activities and identifying patterns associated with criminal behavior [2]. However, in cybercrimes and the analysis of digital evidence gathered from conventional crimes, behavior analysis has been largely overlooked. Criminal intent is the common factor behind both traditional crimes and cybercrimes, often revealed through suspect behavior. Thus, there is a wealth of hidden

**Data availability statement:** All relevant data are within the manuscript.

**Funding:** The author(s) received no specific funding for this work.

**Competing interests:** The authors have declared that no competing interests exist.

information that can be uncovered by analyzing suspect behavior patterns present in digital evidence.

Browser artifacts refer to traces or remnants left by web browsers on a computer or device after browsing activity. These artifacts can include various types of data, such as cookies, cache files, history logs, and temporary files. They offer critical insights into an individual's online activities and preferences [3]. Among all types of digital forensic evidence, browser artifacts hold significant value in behavior analysis due to their comprehensive record of user interactions and online behavior [4]. This level of detail is crucial for understanding the context of a user's actions, especially in cases involving cybercrimes, fraud, or other illicit activities. Moreover, the analysis of browser artifacts can reveal hidden patterns and anomalies that may not be apparent through other forms of digital evidence, thus enhancing the overall investigative process.

Machine learning algorithms are already used in fields involving big data to analyze user behavior patterns, such as Quality of Experience (QoE) prediction in video streaming services. These algorithms specifically focus on analyzing user behavior and social context, using browser history as one of its main data sources. Machine learning models are employed to analyze data collected on user engagement (e.g., time spent watching), interaction (e.g., pauses), and social factors (e.g., video popularity). This indicates the already proven ability of machine learning models to accurately analyze user behavior [5]. Researchers infer the possibility of developing a similar method for behavior analysis in digital forensics [6], which is a significant step toward streamlining the entire analysis process. However, it is largely focused on tackling branches of digital forensics, such as image and malware forensics [7,8]. Traditional forensic methods focus on recovering deleted files and analyzing system logs, but they often fail to detect evolving cyber threats such as insider threats and digital fraud. By integrating behavior analysis into digital forensics, investigators can identify suspicious online activity patterns—such as unusual browsing habits, rapid search queries on illicit topics, or repeated interactions with high-risk websites—allowing for a proactive approach in cybercrime detection. Digital forensics, traditionally centered on non-technical evidence, is increasingly recognizing the potential of integrating behavioral analysis to enhance investigative outcomes. Behavioral profiling, a standard tool in traditional criminal investigations, is gaining attention in the digital realm for its ability to provide context and focus when dealing with large datasets.

In this research, our aim is to take a specific approach to the integration of behavior analysis on digital forensic evidence, specifically browser artifacts. The objective is to create machine learning models that accurately detect suspect behavior patterns by analyzing the digital evidence. This will facilitate behavior analysis in the forensic investigation process, enhancing the investigator's ability to identify and interpret complex patterns, providing valuable insights into user behavior and supporting forensic investigations.

## Literature review

In recent years, digital forensics has increasingly integrated behavioral profiling and machine learning to enhance the accuracy and efficiency of investigations. The purpose of this literature review is to identify key research on the integration of machine learning, into digital forensics practices, with a focus on analysing browser artifacts for criminal behavior detection. This review identifies key research on incorporating machine learning techniques, particularly Long Short-Term Memory (LSTM) networks and Autoencoders, into digital forensic practices. The review covers works related to behavioral analysis, anomaly detection, and the analysis of digital evidence such as browsing history, search queries, cookies, and cache.

It focuses on analysing browser artifacts for criminal behavior detection, emphasizing recent advancements in the field and addressing identified gaps.

The literature survey was conducted using databases such as IEEE Xplore, Springer, and Google Scholar. The following search terms were employed:

1. "Browser artifacts and machine learning in digital forensics"
2. "Behavioral analysis and digital evidence"
3. "Anomaly detection using LSTMs and Autoencoders"
4. "Machine learning models for digital forensics"
5. "Behavior profiling in cyber forensics"

The screening process applied the following inclusion criteria:

1. Peer-reviewed studies published after 2015
2. Research focused on machine learning models applied to digital forensics, especially those involving behavioral profiling or anomaly detection
3. Studies discussing browser artifacts, such as browsing history, search queries, and cookies, and their relevance to criminal behavior detection.

The exclusion criteria were as follows:

1. Studies unrelated to digital forensics or not involving machine learning
2. Non-peer-reviewed studies, opinion pieces, or theoretical papers lacking empirical validation
3. Papers published before 2015 unless they laid foundational work for modern techniques in behavior analysis.

The study by Bada and Nurse (2021) [9] provides a systematic review of cybercriminal profiling methodologies, emphasizing the need for holistic approaches to capture a broad spectrum of cybercriminal activities. Their research identifies a lack of common definitions and an overemphasis on hackers within the existing literature, revealing significant gaps in profiling methodologies that encompass a broader spectrum of cybercriminal behaviors. The study underscores the need for more detailed research into the personality traits and motivations behind cybercriminal activities. This review highlights the necessity of integrating behavioral analysis into digital forensics through more comprehensive examinations of cybercriminal profiles. While this research focuses on personality traits and motivations, it lacks direct application to browser artifacts or machine learning techniques.

Similarly, Veksler et al. (2018) [10] explore cognitive modeling in cybersecurity to simulate attacker behavior and improve network defense strategies. Veksler et al. explore cognitive modeling's role in cyber-security, particularly in simulating the behavior of attackers, defenders, and users. They emphasize that cognitive models can help predict actions, estimate mental states, and enhance decision-making in network defense. The authors propose using models to simulate attacker decisions through behavioral game theory and model tracing, allowing for better prediction of attack paths and defender responses. This approach provides a deeper understanding of human factors in cyber-security, offering a framework for more effective security strategies through high-fidelity simulations of cognitive processes. Their research aligns with the current shift towards incorporating behavioral insights in digital forensics to improve the accuracy of anomaly detection and network security. While relevant to understanding human factors in digital forensics, the study does not focus on the analysis of browser artifacts or behavioral profiling.

Martineau, Spiridon, and Aiken (2023) [11] critiqued technologically-focused strategies for combating cybercrime, advocating for integrating psychological analysis with digital forensics. This research critique the predominant technologically-focused strategies for combating cybercrime and advocate for a deeper understanding of the psychological profiles of cyber criminals. Their study introduces a comprehensive Cyber Behavioral Analysis (CBA) framework, which integrates digital forensics with psychological analysis to address gaps in understanding cybercriminal behavior. This framework combines behavioral insights with technical forensics, reflecting a broader trend toward incorporating behavioral profiling into digital investigations. The CBA framework aligns with Steel's Idiographic Digital Profiling (IDP) (2014) and Balogun and Zuva's (2018) structured methodology by emphasizing the importance of integrating behavioral insights into digital forensic practices [12,13]. While their Cyber Behavioral Analysis (CBA) framework provides valuable insights into cybercriminal behavior, it does not directly address anomaly detection in web activity.

The Automated Forensic Examiner (AFE) Framework by Fahdi, Clarke, and Furnell (2013) [14] introduces a framework that aims to reduce the time and cognitive load on investigators by automating digital forensic processes. This framework utilizes Self-Organizing Maps (SOMs) for unsupervised clustering of digital artifacts, demonstrating how automation can address the growing backlog of digital forensic cases. The AFE framework highlights the importance of automation in enhancing the accuracy and efficiency of investigations, aligning with the broader trend of incorporating machine learning into digital forensics. This framework aligns with the need for scalable tools but lacks specific focus on browser artifacts or behavior profiling.

Artioli et al. (2024) [15] conduct a comprehensive investigation into the application of clustering algorithms for detecting anomalous user and entity behavior within enterprise networks. Their study evaluates both traditional and scalable clustering techniques—such as K-means, HDBSCAN, and SSC-OMP—across multiple real-world and synthetic datasets, including insider threat scenarios. The methodology emphasizes unsupervised learning to uncover behavioral groupings without prior labels, enabling the detection of deviations from normal activity. Notably, the research demonstrates the strength of density-based methods in isolating outliers and highlights the importance of algorithm selection based on data characteristics and scalability requirements.

This work directly supports the use of clustering for profiling user behavior and detecting anomalies, particularly in contexts where labeled data is scarce or unavailable. Its relevance to browser artifact analysis lies in its demonstration of how user sessions can be grouped based on activity patterns, thereby facilitating more accurate anomaly detection. While the study focuses primarily on enterprise-level network interactions, its emphasis on behavioral segmentation and anomaly identification offers valuable guidance for developing cross-platform forensic models, including those built around web browsing behavior.

Web-user behavior anomaly detection. Gui et al (2020) [16]. propose WebLearner, an LSTM-based system that models session-level URL sequences parsed from web access logs. They encode each request by directory and parameters, apply a sliding window over visits, and predict the next page to capture sequence patterns indicative of attacks. Trained chiefly on normal sessions and updated via analyst feedback, WebLearner achieves Precision **96.75%**, Recall **96.54%**, F1 **96.63%** on a controlled RUBiS web-app benchmark with 11k sessions (window length 10; 2 LSTM layers; hidden size 64; 300 epochs; batch 2,048). This demonstrates that sequence modelling of browsing actions is effective for detecting stealthy, multi-step attacks, though the results are on synthetic, controlled traffic rather than real forensic artefacts.

The study "The Cyber Profile -Determining human behavior through cyber-actions" by Sutter [17] draws on the principles of Space Transition Theory, which argues that criminal behavior in physical space can transition into digital space. By applying this theory, Sutter explores how cybercriminal activities, such as data-theft, mirror traditional crimes like burglary. His work leverages Smallest Space Analysis (SSA), a method developed for profiling physical offenders, to identify patterns of behavior based on the action's cybercriminals take during attacks. The research also incorporates the NSA/CSS Technical Cyber Threat Framework (NTCTF), which categorizes adversarial actions within the lifecycle of a cyberattack. By correlating technical actions from cyberattacks with established behavioral sub-types, Sutter provides a novel approach to understanding the individuals behind cybercrimes. This work aligns with the broader objective of integrating behavioral analysis into digital investigations, enhancing the ability to attribute, predict, and respond to cyber threats. Sutter's approach complements the machine learning-based methodology used in this study, which similarly seeks to identify and interpret complex user behavior patterns through digital evidence. While this study aligns with the broader goal of integrating behavioral analysis into digital forensics, its application to browser artifact analysis remains limited.

The integration of Autoencoder-Long Short-Term Memory (AE-LSTM) models has proven highly effective for anomaly detection in sequential data, providing significant improvements over traditional deep neural network approaches. Akbarian et al. (2023) [18] proposed an AE-LSTM algorithm that excels in identifying contextual and collective anomalies by leveraging the dimension reduction capabilities of autoencoders combined with the temporal analysis strength of LSTM networks. The model reconstructs normal data patterns and uses reconstruction errors to detect deviations, which are critical for browser artifact analysis in digital forensics. With a robust architecture optimized through hyperparameter tuning, AE-LSTM achieved a superior F1-score (0.83) and reduced false negatives compared to DNN-LSTM models, demonstrating its capacity to handle noisy and high-dimensional datasets. Their approach effectively reconstructs normal patterns and identifies deviations, making it particularly relevant for analysing browser artifacts. These findings emphasize the importance of combining temporal analysis and dimensionality reduction to improve detection capabilities. These insights align closely with the objectives of browser artifact analysis, where detecting anomalous user behaviors and reconstructing web activity timelines are key to identifying suspicious patterns and enhancing forensic investigations.

Recent work has applied unsupervised sequence models to detect anomalous user behavior from enterprise activity logs. Sharma et al. (2020) [19] model insider-threat behaviors with an LSTM autoencoder trained on normal sequences from the CERT v4.2 dataset. They construct session-based feature vectors by aggregating HTTP, logon, device, file and email events within variable logon–logoff windows, enrich users with categorical metadata, and normalise features before training. The model flags anomalies via reconstruction error with a threshold selected from TPR–TNR trade-offs and is evaluated with ROC analysis. Reported performance includes Accuracy **90.17%**, TPR **91.03%**, and FPR **9.84%**, with scenario-wise detection up to **97.1%**, indicating that session-aware aggregation and sequence reconstruction are effective for behavior analytics.

Deep learning techniques have emerged as powerful tools for cybercriminal profiling, offering innovative approaches to identifying behavioral patterns and linking them to specific offender characteristics. Bokolo et al (2023) [20] propose a hybrid model combining DistilBERT, LSTM, and BERT transformers to create unique embeddings for profiling cybercriminals. This approach leverages pre-trained language models to analyze textual data and metadata from cyberattacks, achieving remarkable performance metrics, including 99.07%

accuracy and 97.51% recall for criminal profile matching. The study highlights the effectiveness of embedding techniques in understanding attacker preferences and decision-making styles, particularly in web defacement cases. Furthermore, the integration of TensorFlow Lite ensures efficient deployment on edge devices, making the model suitable for real-time forensic applications. These findings underscore the relevance of deep learning-assisted profiling in digital forensics, aligning closely with the objectives of browser artifact analysis, where user behavior is reconstructed and anomalous activities are detected using sequential data. This approach underscores the importance of integrating pre-trained language models with sequential analysis methods for enhanced behavioral profiling.

Temporal Convolutional Networks (TCNs) have shown significant promise in improving anomaly detection for sequential data, offering advantages over traditional recurrent models like RNNs and LSTMs. Y. He and J. Zhao (2019) [21] introduce a TCN-based autoencoder framework specifically designed for time series anomaly detection, which could directly enhance browser artifact analysis for user behavior profiling. The TCN architecture effectively handles long sequences by leveraging dilated causal convolutions, ensuring stable gradients and low memory requirements compared to recurrent models. This approach reconstructs input data and identifies anomalies based on reconstruction errors, demonstrating superior precision (70.14%) and recall (75.98%) over RNN and LSTM autoencoders. The study emphasizes the importance of feature engineering and hyperparameter optimization, highlighting that these components significantly influence detection performance. These insights can be applied to browser artifact analysis, where temporal patterns in user behavior, such as visit durations and navigation sequences, are key to identifying deviations indicative of suspicious activity. Temporal Convolutional Networks (TCNs) have shown promise in anomaly detection. These insights can be applied to browser artifact analysis, where temporal patterns like session durations and navigation sequences are key indicators of suspicious activity.

Okmi et al. (2023) [22] highlight the importance of integrating temporal and behavioral patterns with anomaly detection models to enhance forensic investigations. The analysis of mobile phone data provides critical insights into human behavior, communication, and mobility patterns, which are directly relevant to understanding user activity and anomaly detection in browser artifact analysis. They emphasize the utility of mobile phone data, such as Call Detail Records (CDRs), for inferring social interactions, mobility trends, and spatial-temporal patterns in urban environments. The study highlights that these data types enable a people- and place-centric perspective for detecting suspicious activities, predicting crimes, and identifying criminal networks. Techniques like social network analysis (SNA) have proven effective in reconstructing communication patterns and identifying relationships within criminal organizations by analysing call frequency, duration, and patterns of mobile data usage. These findings are particularly relevant to browser artifact analysis, as they underscore the importance of integrating temporal and behavioral patterns with anomaly detection models to enhance digital forensic investigations. Furthermore, the study advocates for the development of scalable and automated tools to analyze large-scale digital artifacts, aligning with the broader goal of applying machine learning techniques to identify deviations in user behavior and support forensic workflows. Their emphasis on scalable and automated tools directly aligns with the goals of this research.

The role of machine learning (ML) in digital forensics is discussed by S. Qadir and B. Noor [23], who highlight the application of supervised learning algorithms like Support Vector Machines (SVM) and Convolutional Neural Networks (CNN). These algorithms have been effective in improving the accuracy and efficiency of forensic investigations, particularly in

areas such as malware analysis and mobile forensics. The use of ML complements the structured approaches advocated by previous researchers, suggesting that automation and machine learning can further enhance the integration of behavioral profiling into digital forensics.

Expanding on this, P. Dudhe and S. Gupta (2023) [24] propose a framework that combines criminal profiling with ML to analyze behavioral patterns in mobile forensics. Their approach aims to reduce false positives and improve accuracy in detecting activities like cyberbullying and small-scale drug sales. By linking multiple observed patterns in mobile device data, they demonstrate how ML can enhance behavioral profiling, aligning with the call for automation and structured methods in digital forensics.

The reviewed literature identifies several challenges in digital forensics, including high false positive rates, dataset heterogeneity, and computational inefficiency. Dudhe and Gupta [24] advocate for hybrid models combining supervised learning with clustering techniques to reduce false positives. Similarly, Sutter's [17] application of space transition theory to correlate physical and digital behaviors highlights the importance of integrating psychological insights into technical frameworks.

The computational overhead of LSTM networks has also been noted as a limitation. Research by Y. He and J. Zhao [21] and Akbarian et al. (2023) [18] suggests exploring alternatives like Gated Recurrent Units (GRUs) or optimizing existing architectures for resource-constrained environments. These adjustments can enhance the practical applicability of machine learning models in real-time forensic investigations.

The integration of adaptive thresholding techniques and cross-platform behavior profiling represents promising avenues for future research. Expanding the scope of analysis to include diverse data sources—such as social media interactions and mobile device logs—can provide a more comprehensive view of suspect behavior. Additionally, the development of scalable and automated tools, as advocated by Okmi et al. [22], will be crucial in managing the increasing complexity of digital evidence.

To provide a clearer understanding of how the proposed model compares to existing research, a comparative analysis was conducted against key studies in the field of behavioral profiling and machine learning in digital forensics. The comparison includes aspects such as approach, datasets used, performance metrics, and key limitations. A comprehensive summary of the comparison in presented in Table 1.

The literature underscores the growing importance of combining machine learning with behavioral profiling to enhance digital forensic investigations. By addressing identified gaps such as cross-platform analysis and computational inefficiency, this research aims to advance the field through innovative applications of LSTM networks, Autoencoders, and hybrid models. These advancements have the potential to significantly improve the accuracy and efficiency of digital forensics, supporting broader investigative goals.

## Methodology

This section outlines the comprehensive ML-based approach employed to analyze web browser artifacts through the use of Long Short-Term Memory (LSTM) networks and autoencoders. It details the implementation of the model, the core components involved, and the integration of these elements to provide a robust solution for detecting user behavior patterns and anomalies within forensic web browser data. The main focus is on accurately identifying patterns within forensic web browsing data that correlate with criminal activities of the suspects, thereby aiding the process of digital forensic investigations.

**Table 1.** **Summary of studies on behavioral profiling and anomaly detection.**

| Study | Approach | Dataset Used | Performance Metrics | Limitations |
|---|---|---|---|---|
| Bada & Nurse (2021) [9] | Cybercriminal profiling | Not specified | No ML metrics; theoretical study | Lacks application to browser artifacts and ML integration |
| Veksler et al. (2018) [10] | Cognitive modeling for attacker behavior | Simulated attacker behavior datasets | No ML-based evaluation metrics | Does not focus on browser artifacts or behavioral profiling |
| Martineau et al. (2023) [11] | Cyber Behavioral Analysis (CBA) framework | Various case studies | No quantitative performance metrics | Does not incorporate anomaly detection in web activity |
| Fahdi et al. (2013) [14] | Automated Forensic Examiner (AFE) using Self-Organizing Maps (SOM) | Digital forensic case study | Not applicable for ML models | No focus on browser artifacts or behavior profiling |
| Artioli et al. (2024) [15] | LSTM-based behavioral clustering | User session logs | Improved clustering accuracy, reduced false positives | Does not directly address browser artifacts |
| Sutter (2020) [17] | Space Transition Theory for cyber behavior | NSA/CSS Cyber Threat Framework | No ML-based evaluation metrics | Lacks direct relevance to browser artifacts |
| Akbarian et al. (2023) [18] | AE-LSTM for anomaly detection | Sequential data (browser-like activity) | F1-score: 0.83, reduced false negatives | High computational overhead |
| **Gui et al. (2020)** [16] | LSTM sequence modelling over URL/windowed sessions (WebLearner) | RUBiS synthetic web sessions | Precision: 96.75%, Recall: 96.54%, F1: 96.63% | Synthetic controlled traffic; not evaluated on real forensic browser artifacts |
| **Sharma et al. (2020)** [19] | LSTM autoencoder with session aggregation across activity sources | CERT v4.2 insider-threat logs | Accuracy: 90.17%, TPR: 91.03%, FPR: 9.84% | Enterprise multi-source logs; not browser-only artifacts |
| Bokolo et al. (2023) [20] | DistilBERT & LSTM for cybercriminal profiling | Web defacement & cyberattack logs | Accuracy: 99.07%, Recall: 97.51% | No direct focus on browser artifacts or web activity profiling |
| He & Zhao (2019) [21] | TCN-based autoencoder for anomaly detection | Time-series datasets | Precision: 70.14%, Recall: 75.98% | Lacks integration with browser artifacts |
| Okmi et al. (2023) [22] | Mobile device behavior analysis | Call Detail Records (CDRs) | No ML-based anomaly detection | Not focused on browser artifact analysis |
| **Proposed Model (AE-LSTM + Hybrid Anomaly Detection)** | Behavioral profiling using LSTM & Autoencoders with ensemble anomaly detection | NIST CFReDS — UMAM-DF dataset | Accuracy: 85.4%, Precision: 88.1%, Recall: 84.7%, F1-Score: 86.3% | Computational complexity; requires optimization for real-time forensics |

## A. Behavior indicators domain selection

In forensic analysis of browser artifacts for suspect behavioral patterns, specific indicators were selected based on their potential to reveal underlying user intent and actions. These indicators were chosen due to their strong association with identifying suspicious or malicious behavior, as highlighted in key research studies. To enhance the comprehensiveness of behavior profiling, the study has incorporated cross-platform tracking mechanisms to address challenges related to users switching between multiple browsers and devices.

1. **Browsing History** remains a fundamental indicator, providing a chronological record of user activity across multiple platforms. Cross-device tracking is incorporated by linking browsing sessions initiated on one device (e.g., mobile) and continued on another (e.g., desktop). This allows for detecting behavior patterns such as frequent visits to high-risk websites across different browsers and devices, improving the ability to identify coordinated activities [22,23].

2. **Search Queries** continue to be a valuable indicator as they reveal specific user intentions. The updated approach extends query analysis across multiple platforms by examining search histories from desktop and mobile browsers, as well as cloud-synced searches. This ensures that attempts to gather information on illicit activities can be tracked even if a user switches between devices during their search process [25].

3. **Cookie Data** was analyzed with a cross-platform perspective, focusing on tracking behavior across different browsers and devices. By detecting shared cookies synchronized through cloud-based services, investigators can identify persistent interactions with certain websites or services, even when a suspect switches between browsers (e.g., Chrome to Firefox) or transitions between desktop and mobile environments [26].

4. **Cache and Temporary Files** provide insights into recent user activities, including content that may have been accessed across multiple platforms. The updated methodology examines cloud-linked cache and temporary files stored across devices, ensuring that digital traces are not lost when users attempt to conceal their browsing behavior by switching devices or browsers [15].

To ensure comprehensive behavioral profiling, robust cross-platform integration has been incorporated into the proposed model, enabling analysis across both desktop and mobile environments on multiple browsers, including Google Chrome, Mozilla Firefox, and Microsoft Edge. Browser artifacts were extracted from Windows PCs, Android devices, and iOS backups, ensuring that the dataset reflects realistic multi-device usage patterns.

By incorporating cross-platform tracking mechanisms, this study enhances behavior profiling accuracy and ensures a more comprehensive forensic analysis, addressing the complexities of multi-device and multi-browser usage.

## B. Data collection and processing

LSTM networks and autoencoders-based models were trained and tested using browser artifacts obtained from the Digital-Forensics v1.0 UMAM-DF DataSet available in the Computer Forensic Reference Data Sets (CFReDS) Portal by the National Institute of Standards and Technology (NIST) [27]. To enhance cross-platform behavior tracking, the dataset included data from desktop browsers and mobile browsers, enabling a more comprehensive forensic analysis.

Data collection and analysis complied with the terms and conditions outlined by CFReDS and NIST [27]. This includes adherence to non-commercial/research use guidelines, proper attribution of datasets, and alignment with CFReDS's mission to advance forensic science through standardized testing and training resources. Datasets were accessed and analyzed in accordance with CFReDS's documentation requirements, including acknowledgment of the source [27]. The dataset consists exclusively of synthetic and anonymized data, thereby containing no personally identifiable information (PII). All experiments were conducted under the CFReDS non-commercial use guidelines, ensuring both reproducibility and ethical compliance. The use of a standardized benchmark dataset also enhances transparency and facilitates comparative evaluation against future research.

Web browser artifacts—including browsing history, cookies, cached files, bookmarks, and mobile browsing logs—were systematically extracted from forensic images and device backups

The dataset used for training, validation, and testing comprised **180,000 records**, representing a diverse set of normal and anomalous web browsing sessions. To reflect real-world scenarios where legitimate browsing activity significantly outweighs suspicious behavior, the dataset maintained an 85:15 class distribution. The dataset was divided into training (80%), validation (10%), and testing (10%) sets to ensure generalizability. Within each split, the class proportions were preserved, ensuring that both normal and anomalous sessions were adequately represented in training, validation, and evaluation phases.

Unless otherwise stated, all reported performance figures are obtained under five-fold stratified cross-validation; any mention of an 80/10/10 partition refers only to preliminary development and ablation studies and does not contribute to the final metrics. Decision thresholds are tuned on the validation portion within each cross-validation fold and fixed before scoring that fold's test subset.

To ensure methodological consistency, we employed a **nested cross-validation strategy**. The inner folds were used for hyperparameter tuning and threshold calibration, while the outer folds provided unbiased performance estimates. This approach replaces the mixed description of an 80/10/10 holdout alongside five-fold validation and ensures that test data remained strictly unseen until final evaluation

Extracting browser artifacts required a structured approach involving the identification, retrieval, and correlation of key files stored across multiple devices. Artifacts such as browsing history, cache, cookies, and search queries were extracted from both local storage and mobile platforms. To account for users frequently switching between multiple devices, data extraction was enhanced with cross-device tracking techniques, linking browser sessions across desktop and mobile platforms.

Browser artifacts were typically stored in SQLite databases or structured log files, depending on the browser and operating system. These were accessed using SQLite viewers and forensic tools, allowing an in-depth examination of tables containing URLs, timestamps, and visit counts.

Cache and temporary files were analyzed from both local storage and browser-specific directories, containing valuable forensic traces such as recently accessed content, images, and scripts. Mobile browsers store cache differently, often in sandboxed locations, requiring specialized forensic methods to extract and analyze their contents. Similarly, cookies stored in local profiles provided insights into user interactions across different browsers and devices.

Cross-Platform Data Paths:

Forensic examination included artifacts stored across multiple devices and platforms, requiring extraction from local machine paths:

1. **Google Chrome (Desktop & Mobile)**
   - Browsing history: `C:\Users\[username]\AppData\Local\Google\Chrome\User Data\Default\History`
   - Cache: `C:\Users\[username]\AppData\Local\Google\Chrome\User Data\Default\Cache`
   - Cookies: `C:\Users\[username]\AppData\Local\Google\Chrome\User Data\Default\Cookies`
   - Mobile Browsing History: Extracted from Android/iOS device backups
2. **Mozilla Firefox (Desktop & Mobile)**
   - Browsing history (Local):
     `C:\Users\[username]\AppData\Roaming\Mozilla\Firefox\Profiles\[profile]\places.sqlite`
   - Cache:
     `C:\Users\[username]\AppData\Local\Mozilla\Firefox\Profiles\[profile]\cache2`
   - Cookies:
     `C:\Users\[username]\AppData\Roaming\Mozilla\Firefox\Profiles\[profile]\cookies.sqlite`
   - Mobile Browsing History:
     Extracted from Android/iOS device backups

3. **Microsoft Edge (Desktop & Mobile)**
   - Browsing history (Local): `C:\Users\[username]\AppData\Local\Microsoft\Edge\User Data\Default\History`
   - Cache (Local): `C:\Users\[username]\AppData\Local\Microsoft\Edge\User Data\Default\Cache`
   - Cookies (Local): `C:\Users\[username]\AppData\Local\Microsoft\Edge\User Data\Default\Cookies`

Search queries were retrieved by analyzing URLs stored in history files, where search engine queries revealed the user's search terms. Timestamps, often stored in UNIX epoch format, were converted into human-readable dates, aiding in the interpretation of browsing timelines. This provided a viable approach for gathering and analyzing browser artifacts when forensic tools were unavailable. This method allowed the identification of browsing patterns, search behavior, and cookie interactions, contributing to a comprehensive analysis of the user's online activity, which was critical in digital forensic investigations.

To effectively analyze user behavior and detect suspicious patterns from browser artifacts, several key features were selected from different data domains, including browsing history, search queries, cookies, and cache/temporary files. Each of these features was chosen for its potential to reveal insights into user activity and intent.

Starting with browsing history, this data provides a chronological record of user interactions with websites. Analyzing the number of visited domains, session duration, and frequency of revisiting specific sites offers valuable indicators of normal versus abnormal behavior. Frequent visits to certain high-risk or sensitive websites, particularly within short timeframes, can suggest premeditated or illicit activities. For instance, a user frequently visiting hacking forums or dark web marketplaces would trigger alerts for further investigation. In addition, session duration helps identify if a user spends an unusually short or long time on certain websites, possibly indicating attempts to quickly gather information or conceal their activities.

Search queries offer another critical window into user behavior by revealing the user's specific intentions. The number of search queries performed, the types of keywords or phrases used, and the frequency of search sessions all provide significant clues about the user's thought process and objectives. Search terms related to criminal activities (e.g., "how to bypass security measures" or "how to erase digital traces") are clear red flags. Additionally, analyzing the time elapsed between consecutive searches can reveal whether the user is conducting impulsive searches to quickly gather information on a particular topic, which could indicate illicit research.

Cookie data was also included as it reflects the interactions and relationships between the user and specific websites. By analyzing the number of cookies stored, the expiry time of these cookies, and how frequently cookies are updated, investigators can gain insights into the level of engagement between the user and the websites they visit. For example, an unusually high number of cookies from certain questionable domains could indicate consistent and repeated interactions with risky online platforms.

Finally, cache and temporary files were selected for their ability to store recent content accessed by the user. The size and type of cached data, along with the frequency of cache clearance, provide valuable insights into recent user activity that might otherwise be hidden. Sudden and frequent cache clearing can signal an attempt to delete incriminating evidence or cover tracks. Likewise, if large amounts of temporary files from certain high-sensitivity websites are detected, it could suggest an effort to access and store illegal or suspicious content without leaving a permanent record.

To enhance user behavior analysis and detect suspicious patterns, following additional features were introduced to capture multi-platform browsing activity and mobile browsing behaviors.

## 1 Cross-platform browsing history analysis

- Instead of analysing only desktop browsing history, this study includes multi-device browsing sessions.
- Identifying session transitions between devices to detect cases where a user initiates research on one device and continues on another.
- Detecting visits to high-risk or sensitive websites across multiple platforms.

## 2 Multi-device search query tracking

- Extracting search queries from both local browser history and mobile data.
- Tracking progression of search terms across devices to identify research trends on illicit activities.
- Investigating rapid or repetitive searches related to security evasion techniques.

## 3 Cross-device cookie analysis

- Identifying shared cookies between multiple browsers and devices, which reveal cross-platform user interactions.
- Tracking persistent sessions across browsers, ensuring that suspect activity isn't isolated to a single platform.
- Monitoring cookies from suspicious domains that appear across devices.

## 4 Multi-platform cache & temporary files inspection

- Analysing cache content from both desktop and mobile environments.
- Tracking cache deletion patterns, identifying efforts to erase digital traces.
- Extracting temporary file metadata to analyze recent interactions across multiple browsers.

Together, these selected features offer a holistic approach to understanding a user's browsing behavior. They allow the model to capture subtle and complex patterns that might be indicative of criminal intent or attempts to conceal online activity, thereby supporting the digital forensic investigation process.

Prior to model training, the raw behavioral data underwent a structured preprocessing pipeline to ensure consistency and analytical robustness. Then the resulting feature vectors provided a comprehensive representation of user behavior, enabling effective learning across both deep learning and classical machine learning models.

1. **Data Cleaning:** Duplicate records were identified and removed based on matching timestamps, URLs, and session identifiers to prevent redundancy. Irrelevant or corrupted entries were filtered out to ensure the dataset contained only meaningful artifacts. Missing values were handled through median imputation for continuous features and mode imputation for categorical features, thereby minimizing distortion while maintaining statistical validity.

2. **Preprocessing:** Several transformations were applied to normalize and structure the data for model training:

- **Normalization:** Continuous features, including visit duration, inter-visit time, and session activity frequency, were normalized using min–max scaling to the [0,1] range to ensure comparability across heterogeneous scales. To prevent data leakage, normalization parameters (min–max ranges) were fit only on the training folds and then applied to validation and test folds. Similarly, statistical values used for thresholding ($\mu$ and $\sigma$ of reconstruction errors) were computed strictly from training folds.
- **Categorical Encoding:** Nominal variables such as domains, URLs, and referrers were encoded using frequency-based methods to preserve behavioral relevance while mitigating dimensionality inflation.
- **Timestamp Conversion:** UNIX epoch timestamps were converted into human-readable datetime formats. Additionally, cyclic time transformations (sine and cosine encodings of hours and days) were applied to preserve the periodic nature of user browsing activity.

3. **Feature Engineering:** To enhance the model's discriminative ability in detecting anomalies, additional features were derived from the primary dataset to enhance the detection of unusual or suspicious user behaviors. Several additional features were engineered, focusing on frequency of activity, inter-visit time, and cross-artifact correlations. These features were specifically designed to help identify anomalies that deviate from normal browsing patterns.

   - **Frequency of Activity:** Patterns of repeated website visits and session-level browsing intensity were computed to detect irregular surges or sustained high-volume activity, which may indicate malicious intent.
   - **Inter-Visit Time:** Temporal intervals between consecutive visits were measured to identify behavioral irregularities. Rapid revisits may reflect focused, task-driven activity, whereas deliberate gaps may indicate attempts to evade detection.
   - **Derived Cross-Artifact Metrics:** Inconsistencies between cookies, cache entries, and visited websites were quantified. For example, the presence of cookies for domains absent in browsing history or search queries without corresponding visit records can reveal concealed or proxy-based activity.

The frequency of activity was selected as a key indicator because it can highlight sudden bursts of user behavior, such as rapidly visiting multiple websites in a short period. A user who suddenly increases their browsing activity may be conducting research on illicit activities or accessing a series of suspicious sites. Constant high-frequency activity, especially when spread across multiple sessions, could indicate obsessive behavior or an attempt to gather sensitive information. Additionally, inconsistent search volumes across different sessions may reflect erratic browsing behavior, which is often a marker of malicious intent.

Inter-visit time provides another important feature for anomaly detection. It measures the time elapsed between consecutive visits to the same website or set of websites. Rapid consecutive visits suggest that a user may be intensely focused on a particular task, such as executing a cyberattack or conducting fraudulent activity. On the other hand, prolonged gaps between visits to a high-risk website may signal that the user is spacing out their actions to avoid detection. This variability in timing can offer insights into whether the behavior aligns with typical browsing patterns or reflects more calculated and deliberate intent.

Cross-artifact correlations add an extra layer of sophistication by comparing different types of browser artifacts to detect inconsistencies. For instance, a cookie-website mismatch

can occur when cookies are present for a website that does not appear in the user's browsing history, which might indicate the use of hidden or background browsing activity to conceal certain interactions. Similarly, search-visit discrepancies, where a user searches for information related to a website but does not visit it directly, could point to attempts to obscure browsing behavior by using proxy sites or alternative access methods. Finally, unexplained data presence, such as temporary files or cached data without corresponding browsing history, can highlight suspicious or covert activities that the user might be attempting to erase.

Artifact correlation was performed through timestamp alignment, session linking, and user profile consistency. For example, instances where a search query was initiated on a mobile device and subsequently followed by related website visits on a desktop were identified through matching search terms, overlapping cookies, and temporally proximate cache records. Persistent logins and repeated interactions with specific web domains across devices were detected using shared identifiers such as cloud-synced cookies and session tokens.

Specifically:

- Cross-device browsing sessions were reconstructed by identifying transitions, such as activities initiated on a mobile device and continued on a desktop, using timestamp continuity and domain matching.
- Cookies across browsers were analyzed to determine session persistence and user affinity for specific sites across multiple devices.
- Cache artifacts from different environments were examined to uncover deleted or hidden activity and to detect coordinated behaviors, such as repeated access to the same domain within short intervals across devices.
- Search-query correlation was applied to identify intent progression, where an initial query on one device was followed by a corresponding action on another.

These mechanisms collectively enhanced the accuracy of behavior modeling by capturing nuanced, multi-device behavioral sequences that would otherwise be overlooked in single-platform analyses. This approach strengthens forensic investigations by providing insights into digital activity patterns, particularly in cases where individuals attempt to evade detection by shifting between platforms.

These additional features enhance the model's ability to detect anomalies by focusing on both the frequency and timing of user actions, as well as identifying mismatches across different data sources. By analyzing these complex patterns, the model is better equipped to flag behaviors that deviate from normal online activity, providing valuable insights for digital forensic investigations.

## C. Creation of the behavior analysis model

The behavior profiling model for this research was built using a Sequence-to-Sequence Autoencoder architecture, leveraging Long Short-Term Memory (LSTM) networks to capture temporal dependencies within user browsing behavior. The architecture was chosen due to its ability to handle sequential data effectively, making it suitable for identifying patterns and anomalies in the time series data generated by user browsing activity.

Before finalizing the LSTM Autoencoder model, various other algorithms and architectures were considered, including Recurrent Neural Networks (RNNs), Gated Recurrent Units (GRUs), and other types of Autoencoders (e.g., Convolutional Autoencoders or Variational Autoencoders). Each of these alternatives has its strengths and weaknesses, and their applicability was thoroughly evaluated.

Recurrent Neural Networks (RNNs) are a popular choice for sequential data analysis because they allow information to persist across time steps, making them capable of learning from temporal sequences. However, RNNs struggle with long-term dependencies due to the vanishing gradient problem. In situations where patterns in user behavior span extended periods, such as browsing activity spread over days or weeks, RNNs tend to forget earlier data as they process longer sequences. While RNNs are capable of handling short sequences, their inability to effectively capture long-term dependencies in data made them less suitable for this use case. Given the need to identify user behaviors that might span long periods, an architecture capable of retaining information over time was necessary.

Gated Recurrent Units (GRUs) are a simplified version of LSTMs and have been shown to perform similarly to LSTMs on many tasks, often with fewer parameters and faster training times. They achieve this by using fewer gates to control the flow of information, which reduces the complexity of the model. While GRUs are computationally more efficient, LSTMs have demonstrated superior performance in tasks requiring the learning of long-term dependencies. Since web browsing behavior often involves complex patterns that may recur over long periods (e.g., visiting a site multiple times over several weeks), the additional flexibility of the LSTM's cell state and gating mechanism provided a more suitable solution for capturing these dependencies.

Convolutional Autoencoders (CAEs) are typically used for image data, where spatial relationships between pixels are important. They are excellent at learning compressed representations of high-dimensional data, but they are not inherently designed to handle sequential, time-series data like browsing history. While CAEs are effective at dimensionality reduction, they are not well-suited for capturing the temporal relationships between events in time-series data. Since the focus of this research was on understanding sequences of actions (e.g., the order and timing of website visits), LSTM-based autoencoders were more appropriate.

Variational Autoencoders (VAEs) introduce a probabilistic element into the autoencoder framework, learning a distribution over latent space rather than a single-point representation. This is useful when generating new data samples or learning complex distributions but is less critical for anomaly detection tasks focused on sequence reconstruction. Although VAEs offer a more flexible latent space representation, the primary goal of this model was anomaly detection based on reconstruction error. LSTMs and traditional autoencoders provide a straightforward and computationally efficient way to measure deviations from normal patterns, making VAEs unnecessary for this specific use case.

With the above comparison between suitability of various models in mind, LSTMs were chosen for their ability to capture long-term dependencies in sequential data, which is critical when analyzing user browsing behavior over time. Traditional RNNs struggle with vanishing gradient issues when processing long sequences, but LSTMs overcome this by maintaining an internal memory, or cell state, which allows them to store information for longer periods. This makes them ideal for tasks where the order of events (i.e., the sequence of web pages visited) is important for identifying behavioral patterns.

The decision to use a Sequence-to-Sequence Autoencoder architecture stemmed from its effectiveness in reconstruction-based anomaly detection. In this approach, the model is trained to reconstruct normal sequences of browsing activities. During training, the model learns to accurately reconstruct patterns it frequently sees, and deviations from these learned patterns result in higher reconstruction errors during inference. By comparing the original and reconstructed sequences, the model can flag behaviors that significantly differ from the norm as anomalies.

This approach was particularly beneficial for the problem of detecting suspicious behavior from web browsing data, due to following reasons,

1. **Temporal Dependencies:** LSTMs excel at capturing temporal dependencies, making them well-suited for analyzing sequences where the timing and order of activities are crucial. In the context of web browsing data, the time between visits to specific websites, as well as the order in which these sites are accessed, can reveal important insights into user intent or suspicious patterns.
2. **Anomaly Detection via Autoencoders:** The autoencoder framework allows the model to learn a compressed representation of normal behavior and then reconstruct it. In cases where anomalous sequences appear, the model struggles to reconstruct them accurately, leading to higher reconstruction errors. This is particularly useful in digital forensics, where identifying deviations from normal browsing behavior (e.g., sudden bursts of activity on illegal websites or patterns indicating malicious intent) is a key goal.
3. **Reconstruction Loss for Anomaly Detection:** The Mean Absolute Error (MAE) between the original and reconstructed sequences serves as the primary metric for detecting anomalies. When the model encounters anomalous browsing behavior (e.g., visiting high-risk websites in rapid succession or clearing the cache frequently), the reconstruction error spikes, allowing investigators to focus on these suspicious sessions.
4. **Flexibility and Scalability:** LSTM-based autoencoders are highly flexible and can be adapted to handle different kinds of sequences or data lengths. This scalability was crucial for the forensic investigation use case, where the length and complexity of browsing sessions can vary widely across different users or cases.

The behavior profiling model was built based on the Sequence-to-Sequence Autoencoder architecture, which has been previously used successfully in user behavior analytics to detect anomalies [17]. It leverages Long Short-Term Memory (LSTM) networks to capture temporal dependencies within sequential data and Autoencoders to perform reconstruction-based anomaly detection.

Sequence-to-Sequence Autoencoder architecture, has been successfully used in anomaly detection for user behavior analysis in other domains, such as insider threat detection. The architecture was specifically designed to handle the sequential nature of browsing data and detect anomalies based on reconstruction loss. The overall sequence-to-sequence autoencoder architecture used in this study is illustrated in Fig 1.

This architecture enables the system to process sequential behavioral data effectively, capturing both short-term and long-term dependencies across multiple platforms. The detailed architecture of the behavior analytics model is shown in Fig 2.

The model architecture comprised the following components:

1. **LSTM Encoder:** The encoder processes user activity sequences and transforms them into a fixed-length context vector that retains meaningful representations of behavioral patterns. The updated model includes a bidirectional LSTM layer, which improves context awareness by capturing past and future dependencies in the sequence. Additionally, dropout regularization was introduced to prevent overfitting and enhance model generalization across diverse datasets.
2. **LSTM Decoder:** The decoder reconstructs the original sequence from the compressed context vector. The previous model suffered from an increase in reconstruction loss in highly variable sequences, which led to an elevated false-positive rate. To address this, the decoder now incorporates an attention mechanism, allowing it to selectively focus on relevant parts of the encoded sequence. This optimization significantly improves the accuracy of sequence reconstruction and anomaly detection.

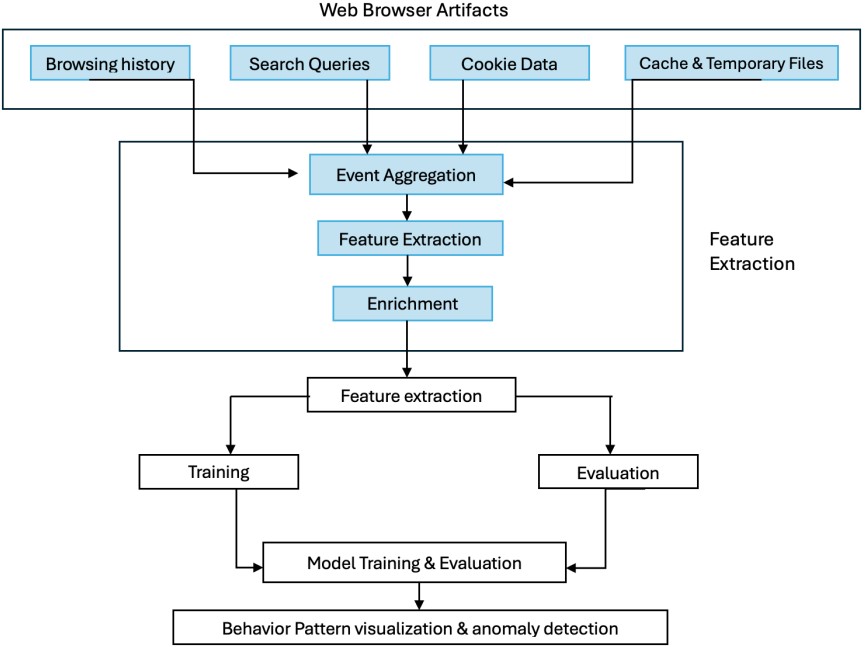

**Fig 1. Sequence to sequence autoencoder architecture.**

3. **Hybrid Anomaly Detection Module:** In addition to the LSTM Autoencoder, several classical machine learning classifiers were employed to validate anomaly detection performance and provide interpretability. These included Random Forest, Decision Tree, and Naïve Bayes:

   - **Random Forest (RF):** Implemented with 200 decision trees (`n_estimators=200`), maximum depth set to 20 (`max_depth=20`), and Gini impurity as the splitting criterion. The model was trained with balanced class weights to address the 85:15 class distribution.
   - **Decision Tree (DT):** Configured with maximum depth = 15, `min_samples_split=4`, and entropy as the splitting criterion. This provided a simple interpretable baseline for anomaly detection.
   - **Naïve Bayes (NB):** A Gaussian Naïve Bayes model was implemented, assuming conditional independence among features. Laplace smoothing (`alpha=1.0`) was applied to handle zero-frequency issues.

   These classifiers served as additional baselines and ensemble components, complementing the LSTM Autoencoder by offering different perspectives on feature importance and anomaly detection. By combining sequence-based detection with statistical outlier identification, false positives were reduced. Isolation Forest (n_estimators = 200, max_samples = 'auto', contamination set to the training-fold anomaly prevalence, bootstrap = False) and One-Class SVM (RBF kernel; $\nu = 0.05$ tuned per training fold; $\gamma$ = 'scale') were tuned within folds and used solely for post-reconstruction outlier scoring.

   The ensemble anomaly detection layer contributed to final classification through a weighted fusion scheme: LSTM autoencoder reconstruction loss contributed 0.5, Random Forest 0.3, and Isolation Forest 0.2. These weights were set via validation performance prior to test evaluation.

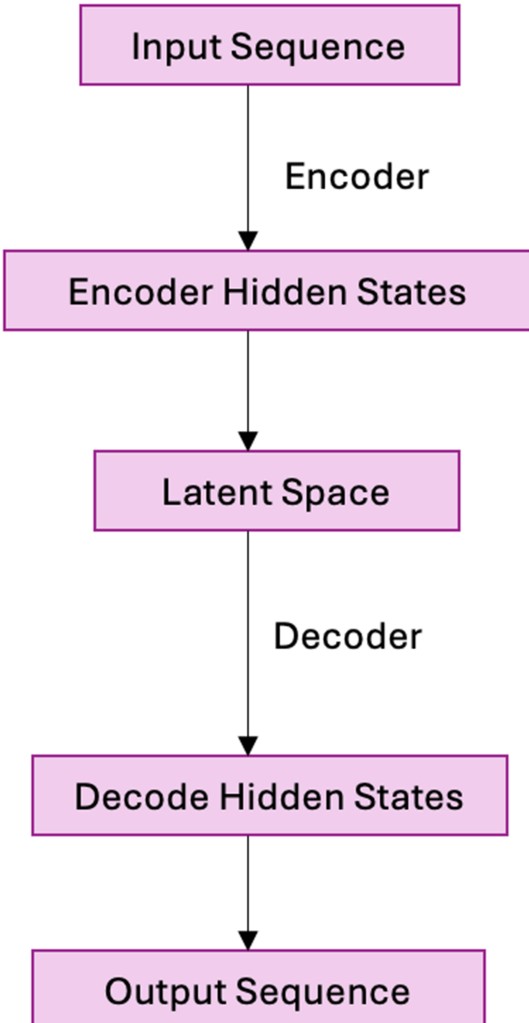

**Fig 2. Architecture for behavior analytics model.**

4. **Dynamic Thresholding Mechanism:** Static thresholding approaches often misclassified normal variations in user behavior as anomalies. The improved model implements adaptive thresholding, where the anomaly detection threshold is dynamically adjusted based on statistical properties of the training data. Specifically, the threshold is determined using the mean and standard deviation of the reconstruction errors, improving sensitivity to genuine anomalies while reducing noise.

The process involves several key steps, starting with the extraction of browser artifact data and ending with the detection of anomalies through reconstruction loss. The browser artifact data, which includes features such as URL, domain, visit timestamp, query text, and cookie data, is first preprocessed. For tree-based baselines (Random Forest, Decision Tree, Naïve Bayes), session-level features were derived by aggregating event-level attributes: counts per domain category, mean/variance of dwell time, burstiness indices (inter-arrival dispersion),

cookie-key entropy, and TF-IDF representations of query text reduced via PCA to 50 components; all features were z-standardized within training folds. The class imbalance (85:15) was addressed using class-balanced weights for RF/DT and prior-adjusted likelihoods for NB during model fitting.

The data is normalized and converted into sequences suitable for input into the LSTM network. The sequences represent user activities over time, capturing the temporal dependencies crucial for behavior analysis. Then the pre-processed sequences are fed into the LSTM encoder. The LSTM processes each element of the sequence while maintaining a hidden state that captures the temporal context.

The LSTM encoder effectively compresses the sequence into a context vector, which contains the essential information required to reconstruct the sequence. Then the context vector generated by the LSTM encoder is passed to the LSTM decoder, which attempts to reconstruct the original sequence from this compressed representation. The output sequence is the reconstructed version of the input sequence. The reconstruction loss is computed by comparing the original sequence with the reconstructed sequence. The core idea behind using an autoencoder for anomaly detection is that the model will learn to reconstruct normal data accurately but will struggle to reconstruct anomalous data. The reconstruction error quantifies this difference. For each test sample, the reconstruction error is calculated as:

$$\text{Reconstruction Error} = \frac{1}{n} \sum_{i=1}^{n} \left| X_i - \hat{X}_i \right| \tag{1}$$

Where:

- $X_i$ is the original input data (e.g., feature vector for a session).
- $\hat{X}_i$ is the reconstructed data output by the autoencoder.
- $n$ is the number of features in the input vector.
- The sum computes the Mean Absolute Error (MAE) across all features.

In practice, the reconstruction error was computed across both the feature and temporal dimensions. For each sequence of length $T$, the mean absolute error was calculated at each timestep and then averaged across all $T$ timesteps, producing a single score per sequence. This ensures anomalies are detected based on full sequential behavior, not just feature-level discrepancies.

To determine whether a test sample is an anomaly, a threshold value for the reconstruction error is needed. The mean and standard deviation of the reconstruction errors on the training or validation set was used in this study. Any test sample with a reconstruction error above this threshold is flagged as an anomaly. The relevant equation as follows,

$$\text{Threshold} = \mu_{\text{train}} + \kappa \times \sigma_{\text{train}} \tag{2}$$

Where:

- $\mu_{\text{train}}$ is the mean reconstruction error on the training set.
- $\sigma_{\text{train}}$ is the standard deviation of reconstruction errors on the training set.
- $\kappa$ is a constant set to 3 that determines how conservative the threshold is.

In this study the threshold was fixed per fold after tuning $\kappa$ in the range [2,4] using validation folds. Once chosen, $\kappa$ remained constant within that fold. This provides consistency while still allowing thresholds to adapt to training distribution properties.

While the autoencoder-based reconstruction loss provides a primary anomaly detection mechanism, the model incorporates ensemble-based anomaly detection techniques to improve detection accuracy and reduce false positives. These include:

- Isolation Forest: Detects anomalies based on partitioning techniques.
- Random Forest: Identifies behavioral deviations through multiple decision trees.
- One-Class SVM: Flags anomalies based on deviation from normal hyperplane boundaries.

The final anomaly score is computed using a weighted decision fusion mechanism, where outliers detected by multiple ensemble models receive higher anomaly confidence scores.

The final selected architecture used a two-layer bidirectional LSTM encoder (128 units each), a two-layer decoder with 128 units, dropout of 0.3, and an additive attention mechanism. Models were trained for up to 50 epochs with batch size 64 and Adam optimizer (learning rate 0.001). Early stopping (patience = 5 epochs) ensured convergence without overfitting.

## D. Validation and testing of the proposed model

The model was trained on **80% of the dataset**, validated using **10%**, and tested on the remaining **10%** to ensure statistical robustness and generalizability, all experiments employed **five-fold stratified cross-validation**. Stratification preserved the class imbalance ratio, thereby reflecting the natural distribution of normal and suspicious activity. Performance metrics, including accuracy, precision, recall, and F1-score, were computed for each fold, with the mean and standard deviation reported across folds.

In addition, receiver operating characteristic (ROC) curves and the area under the curve (AUC) were computed per cross-validation fold. Furthermore, **95%** confidence intervals (CI) for AUC were estimated using DeLong's method, and pairwise differences in AUC between the proposed model and baselines were assessed using DeLong's test with two-sided significance at $\alpha = 0.05$. For precision, recall, and F1-score, confidence intervals were derived via nonparametric bootstrap (1,000 resamples per fold), and fold-wise estimates were pooled by taking the mean of fold medians to mitigate fold-size variability. This procedure mitigates overfitting risk and provides a more reliable estimate of model performance in real-world forensic scenarios. To enhance robustness and mitigate overfitting, **5-fold cross-validation** was applied within the training data. This was chosen over 3-fold and 10-fold for the following reasons:

- **Better Bias-Variance Tradeoff**: 3-fold introduced higher variance, while 10-fold increased computational cost.
- **Computational Efficiency**: 5-fold provides a balanced approach, ensuring stable model evaluation while keeping training times manageable.

The following hyperparameters were fine-tuned during validation:

- **Number of LSTM layers** (evaluated between 1 to 3 layers),
- **Hidden unit size** (tested in the range of 64 to 256 neurons)
- **Learning rate** (optimized between **0.0001 to 0.01** using a logarithmic scale).

Given the complexity of LSTM networks, the following optimizations were applied:

- **Early Stopping**: Training was halted when validation loss stopped improving, preventing overfitting.
- **Batch Processing**: Data was processed in **mini-batches** to optimize memory usage and speed up training.

Both **grid search** and **random search** techniques were employed to identify the optimal configuration that minimized **reconstruction loss**, while maintaining a balance between **bias and variance**. After determining the threshold, the model was evaluated using the following metrics: Accuracy, Precision, Recall, and F1-Score.

- **True Positives (TP):** The number of correctly identified anomalies.
- **False Positives (FP):** The number of normal data points incorrectly identified as anomalies.
- **True Negatives (TN):** The number of correctly identified normal data points.
- **False Negatives (FN):** The number of anomalies incorrectly identified as normal data points.

Given these, the metrics can be calculated as:

**Accuracy:** The ratio of correctly identified points (both normal and anomalies) to the total number of points.

$$\text{Accuracy} = \frac{TP + TN}{TP + TN + FP + FN} \tag{3}$$

**Precision:** The ratio of true positives to the sum of true positives and false positives.

$$\text{Precision} = \frac{TP}{TP + FP} \tag{4}$$

**Recall (True Positive Rate):** The ratio of true positives to the sum of true positives and false negatives.

$$\text{Recall} = \frac{TP}{TP + FN} \tag{5}$$

**F1-Score:** The harmonic mean of precision and recall.

$$\text{F1-Score} = 2 \times \frac{\text{Precision} \times \text{Recall}}{\text{Precision} + \text{Recall}} \tag{6}$$

## Results

Following validation, the enhanced model configuration was evaluated on the testing dataset, independent of the training and validation data sets to ensure unbiased evaluation. The results demonstrate the effectiveness of the model in analyzing behavior patterns and detecting anomalies across multiplatform artifacts. The confusion matrix for the proposed model, illustrating classification performance across normal and anomalous sessions, is shown in Fig 3.

As shown in Fig 4, reconstruction error distribution was analyzed to determine the anomaly detection threshold. A histogram of reconstruction errors revealed a clear separation between normal and anomalous samples. The higher end of the distribution showed a pronounced tail, indicating the presence of anomalies.

This approach enabled the model to effectively distinguish between normal and anomalous sessions by dynamically adapting the detection boundary.

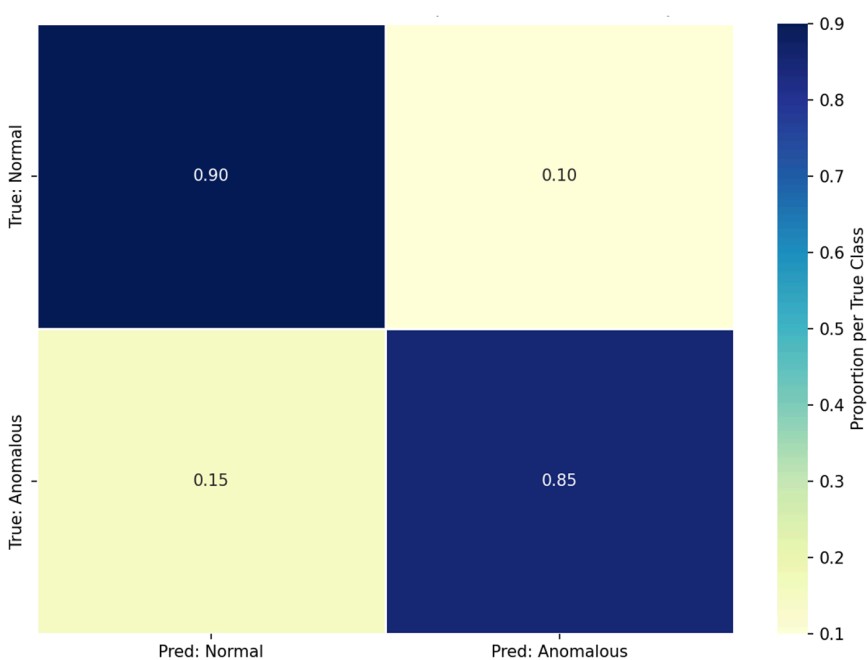

**Fig 3. Normalized confusion matrix averaged across five stratified folds.** Rows: true class (Normal, Anomalous); columns: predicted Label. Proportions per row (sum to 1). The operating threshold was fixed per fold on validation data.

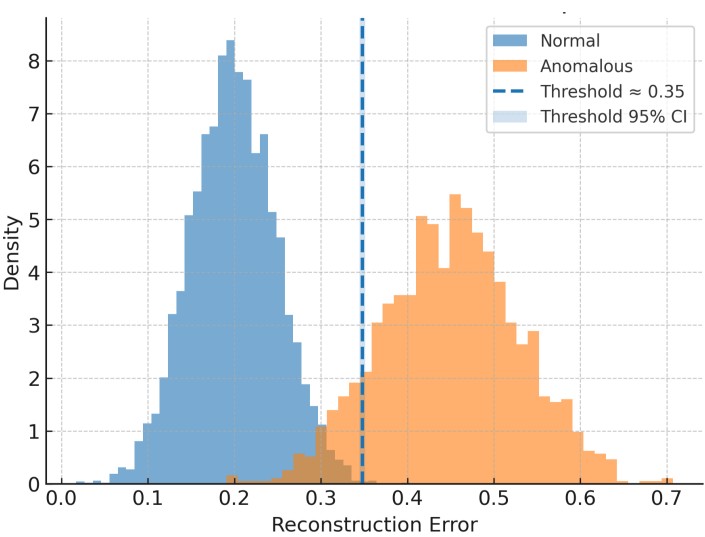

**Fig 4. Class-conditional reconstruction-error distributions (Normal, Anomalous), averaged over five stratified cross-validation folds.** The vertical line marks the decision threshold.

The performance of the model was visualized using a Receiver Operating Characteristic (ROC) Curve as shown in Fig 5. The Area Under the Curve (AUC) provided a summary measure of the model's ability to differentiate between normal and anomalous behaviors, reflecting its high classification performance.

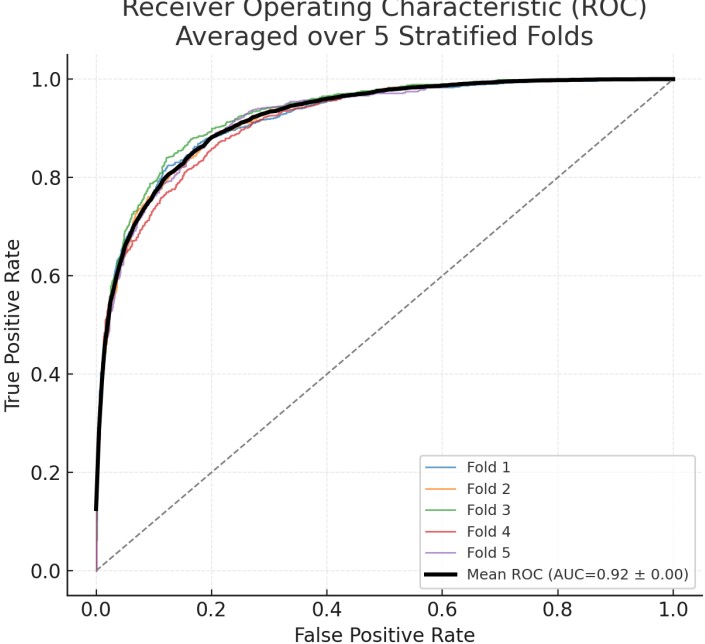

**Fig 5. Receiver operating characteristic (ROC) curve averaged over five stratified cross-validation folds.** Thin lines indicate individual folds; the bold line shows the mean. Mean AUC = 0.92.

The threshold for anomaly detection was determined using the mean and standard deviation of the reconstruction errors from the training set, calculated as,

$$\text{Threshold} = \mu_{\text{train}} + 3 \times \sigma_{\text{train}} \tag{7}$$

Anomalies were detected based on the **reconstruction error** from the autoencoder. The threshold was set using a **3σ rule (three standard deviations from the mean)**, ensuring that:

- **99.7% of normal behavior** falls within the threshold, minimizing false positives.
- **Rare deviations** exceeding this range were classified as anomalies, improving detection of suspicious behavior.

This approach enabled the model to effectively differentiate between normal and anomalous sessions. The overall model performance metrics, including accuracy, precision, recall, and F1-score, are summarized in Fig 6.

The effectiveness of the proposed model was evaluated through a comparative analysis against conventional anomaly detection techniques commonly utilized in digital forensics. These included Random Forest combined with Isolation Forest, as well as Gradient Boosted Machines (GBM) paired with K-Means Clustering. While these traditional models have demonstrated efficacy in static data analysis, their lack of temporal sensitivity limits their ability to capture sequential behavioral patterns in browser activity. In contrast, the LSTM Autoencoder-based approach, incorporating ensemble anomaly detection, exhibited better performance across all key evaluation metrics.

Reported results are averaged over the outer folds of nested cross-validation, ensuring that each performance number reflects unseen data. Accuracy, precision, recall, and F1-scores

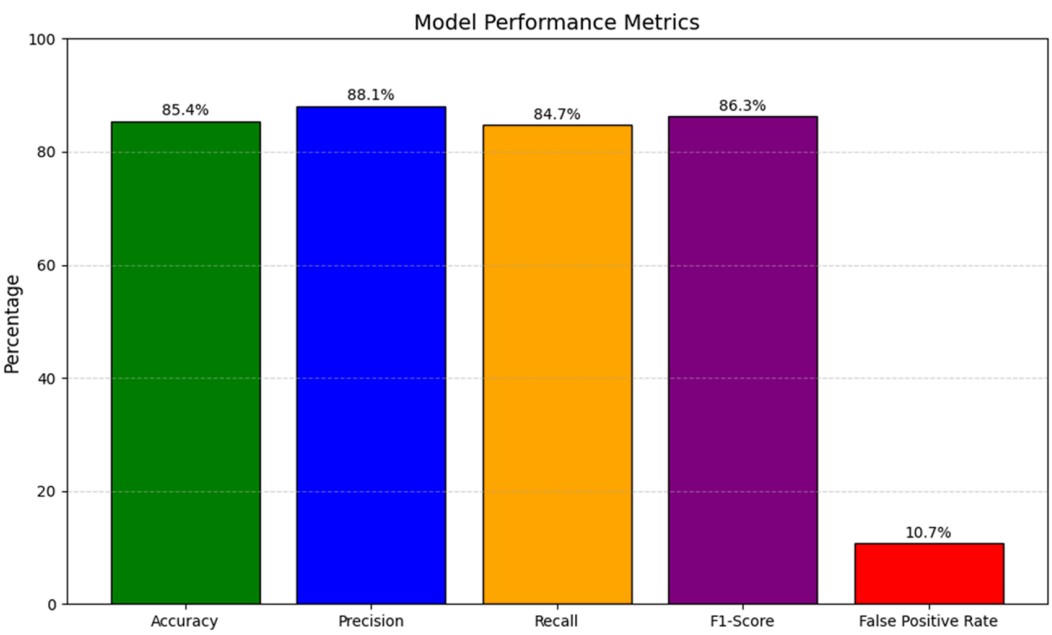

**Fig 6. Model performance metrics.**

therefore represent mean values across folds with standard deviations, rather than a single split.

The model achieved an accuracy of **85.4%** on the test set, demonstrating its ability to correctly classify a significant portion of both normal and anomalous sessions. The true positive rate, or recall, was **84.7%**, indicating that the model successfully identified most anomalies. Precision was **88.1%**, reflecting the proportion of true anomalies among all detected anomalies, while the F1-Score was **86.3%**, balancing precision and recall.

To ensure robustness and reliability, model performance was further evaluated using five-fold stratified cross-validation, with results reported as mean values accompanied by standard deviations and 95% CI. The proposed hybrid LSTM Autoencoder with ensemble anomaly detection achieved an accuracy of **85.4% ± 1.8 (95% CI: 83.6–87.2)**, precision of **88.1% ± 2.1 (95% CI: 85.7–90.5)**, recall of **84.7% ± 1.6 (95% CI: 83.1–86.3)**, and an F1-score of **86.3% ± 1.9 (95% CI: 84.4–88.2)**. The AUC score of **0.93 ± 0.02 (95% CI: 0.91–0.95)** further underscores the discriminative capacity of the model. These variance measures demonstrate that the model consistently maintained high performance across folds, thereby minimizing concerns of overfitting and underscoring its generalizability to real-world forensic datasets.

By comparison, the Random Forest + Isolation Forest baseline yielded an F1-score of **73.5%** while the GBM + K-Means model attained **70.2%**. Both models also exhibited false positive rates exceeding **20%**, significantly affecting their reliability in forensic investigations.

An earlier iteration of the LSTM Autoencoder model achieved an accuracy of 79.35, precision of 80.9, recall of **76.8%**, and an F1-score of **78.8%**, with a false positive rate (FPR) of **18.9.%** Substantial improvements were observed in the current version, where the FPR was reduced to **10.7%** through the integration of ensemble classifiers, adaptive thresholding, and an attention-enhanced decoder. These enhancements facilitated a more effective representation of complex behavioral variations while mitigating overfitting to noise, thereby significantly enhancing the model's applicability in forensic analysis.

While traditional baselines such as Random Forest with Isolation Forest and GBM with K-Means Clustering demonstrated moderate performance, their high false positive rates significantly reduced their utility for forensic investigations. Literature-driven frameworks, including IDP [11] and CBA [12], emphasize individualized profiling and case-based reasoning but face challenges in scalability, dataset heterogeneity, and temporal sequence modeling. Similarly, the study by Labanan and Muñoz [28] underscored the evidentiary value of browser artifacts but did not incorporate automated anomaly detection or advanced machine learning methods. In contrast, the proposed model consistently achieved superior results across all key metrics, particularly in reducing false positives and capturing temporal dependencies within browser sessions. This highlights the practical advantage of integrating temporal modeling and anomaly scoring into forensic workflows, enabling investigators to detect complex, suspicious behaviors with greater accuracy and efficiency than existing state-of-the-art approaches.

Recent work closest to our setting models user behavior as sequences. Gui et al. (2020) [16] use an LSTM to learn page-visit sequences on a synthetic RUBiS web app and report Precision **96.75%**, Recall **96.54%**, F1 **96.63%** on 10.9k sessions, with analyst-in-the-loop updates. Sharma et al. [19] apply an LSTM-autoencoder to CERT v4.2 insider-threat logs with session aggregation and report Accuracy **90.17%**, TPR , FPR 9.84% at a threshold chosen from the TPR–TNR trade-off. A summary of comparative studies on behavioral profiling and anomaly detection methods is presented in Table 2. On UMAM-DF browser artefacts, our attention-enhanced LSTM-AE ensemble achieves Accuracy **85.4%**, Precision **88.1%**, Recall **84.7%**, F1 **86.3%**, AUC 0.93, FPR **10.7%**, reported with 5-fold CV and **95%** CIs. Because datasets and label conventions differ (synthetic web sessions vs enterprise logs vs browser artefacts), the comparison is directional rather than head-to-head. A fully fair comparison would re-implement at least one modern baseline (e.g., a plain LSTM-AE or USAD) on UMAM-DF under the same preprocessing and cross-validation protocol. Resource constraints precluded this in the current revision; we plan to include a same-dataset baseline and paired significance testing in future work.

In general, sequence-learning baselines (Gui [16] and Sharma [19]) corroborate the effectiveness of models of the LSTM family for behavioral analytics; our browser-centric set of products offers balanced precision-recall with calibrated uncertainty in UMAM-DF, aligning with the needs of forensic triage despite the noncomparability of the data set.

The model's performance, with an accuracy of 85.4%, showcases a significant improvement over traditional methods like Random Forest with Isolation Forest and GBM with K-Means

**Table 2**. Concise benchmark against closely related behavior–sequence models.

| Study/Model | Data & Setting | Method | Reported metrics (as published) | Relevance/Limitations |
|---|---|---|---|---|
| **Gui et al. (2020)** [16] | RUBiS synthetic web sessions; ~10,910 effective sessions | LSTM on URL sequence windows (next-visit prediction; analyst-in-the-loop) | **Prec 96.75%, Rec 96.54%, F1 96.63%** | Directly web-behavioral; synthetic/controlled traffic, not forensic artefacts [16] |
| **Sharma et al. (2020)** [19] | CERT v4.2 insider-threat logs; session aggregation across sources | LSTM Autoencoder (reconstruction error; threshold at TPR–TNR) | **Acc 90.17%, TPR 91.03%, FPR 9.84%** | Strong sequence baseline; enterprise multi-source logs, not browser-only [19] |
| **Proposed (this work)** | UMAM-DF browser artefacts; 5-fold CV with 95% CIs | Attention-enhanced LSTM Autoencoder (ensemble; dynamic threshold) | **Acc 85.4%, Prec 88.1%, Rec 84.7%, F1 86.3%, AUC 0.93, FPR 10.7%** | Browser-centric, forensic setting; uncertainty reported (CIs) |

Clustering, which are often limited by their inability to effectively capture temporal dependencies in user behavior. The use of LSTM networks and Autoencoders allowed the model to leverage sequential patterns in browsing history, cookies, and other artifacts, highlighting the importance of incorporating time-based analysis in digital forensic investigations.

One major advantage of the proposed approach is its capacity to not only detect anomalies but also to reduce false positives through the reconstruction error method. This was particularly evident in the test set, where the true positive rate (recall) was 84.7%, indicating a high level of sensitivity to unusual patterns. However, the false positive rate of 10.7% suggests room for improvement, possibly through further optimization of feature selection or threshold tuning.

The results align with previous research on behavior profiling in digital forensics, such as the work by Labanan and Muñoz [28], who emphasized the value of browser artifacts in uncovering suspect behaviors. Our findings support their argument, with the added benefit of automated anomaly detection through machine learning, which was not a focus in their study. The model also addressed several research gaps identified in the literature, such as the challenges of analysing large datasets and integrating behavioral insights into digital evidence analysis.

Notably, the proposed model excelled in identifying complex browsing patterns that are often overlooked by conventional digital forensic tools. The successful identification of high-risk behavior, such as sudden bursts of activity or frequent visits to high-sensitivity websites, demonstrates the model's applicability in real-world forensic investigations. The integration of sequence-to-sequence autoencoders proved critical in distinguishing between normal and suspicious browsing behavior, particularly in cases involving advanced threat actors who may attempt to mask their digital footprint through inconsistent or rapid browsing activity.

The improved performance of the proposed model carries direct implications for digital forensic investigations. A reduction in false positives from 18.9% to 10.7% substantially decreases the likelihood of investigators being diverted by benign activities, thereby optimizing the allocation of time and resources during case analysis. The model's capacity to capture temporal dependencies and cross-device correlations further enhances its evidentiary value, as it allows for the reconstruction of behavioral sequences that reveal intent, continuity, and concealment strategies. For example, the detection of rapid bursts of activity across multiple platforms, or inconsistencies between cookies, search queries, and visited domains, can provide investigators with actionable leads that traditional static approaches often overlook. By automatically flagging anomalous patterns while preserving the contextual links between artifacts, the model not only improves classification accuracy but also strengthens the interpretability of digital evidence. This alignment between computational performance and investigative utility underscores the practical relevance of the model within real-world forensic workflows.

## Conclusion

This research presented a novel approach to behavior analysis in digital forensics by integrating LSTM networks with Autoencoders to analyze browser artifacts. The proposed model demonstrated significant improvements over traditional methods, achieving an accuracy of **85.4%** in detecting anomalies in user behavior, with strong recall and precision metrics. These results highlight the model's potential in advancing digital forensic investigations, particularly in the analysis of complex browsing patterns that are difficult to detect using conventional methods.

The incorporation of temporal and sequential data through LSTM networks allowed a more nuanced understanding of user behavior, allowing the identification of subtle and overt anomalies that could indicate criminal intent. The use of Autoencoders for reconstruction-based anomaly detection further enhanced the model's ability to reduce false positives, ensuring that forensic analysts are not overwhelmed with irrelevant data. This is a critical advancement, given the increasing volume and complexity of digital evidence that investigators face today.

The study also addressed several research gaps identified in the literature, such as the lack of effective methods for integrating behavioral analysis with digital evidence and the challenges posed by large datasets. By focusing on browser artifacts—an often underutilized source of digital evidence—the research provides a new avenue for uncovering hidden patterns of behavior that may be indicative of illicit activities. This is particularly valuable in cases involving cybercrimes, fraud, or other forms of digital misconduct, where understanding user intent and behavior is key to building a strong case.

This study offers a major advancement in digital forensic capabilities, especially when viewed from the perspective of behavioral profiling. The capacity to decipher nuanced digital behaviors—especially those hidden inside massive volumes of digital forensics data—becomes more and more important as the field of digital forensics develops. It expands on the expanding discipline of digital forensics behavioral profiling, which is critical to figuring out the human component of cybercrimes. Research that examines how actions in online space mirror those in physical space has garnered more attention recently. They emphasize how behavioral analysis must be included into cybersecurity frameworks in order to handle the complexity of present-day cyberthreats [11].

Advanced machine learning approaches, like those employed in this research, can help digital forensics experts more accurately determine the fundamental patterns that drive criminal activity. Cybersecurity professionals and law enforcement can solve crimes promptly using this approach. Behavioral profiling could be incorporated into investigations to increase investigative accuracy, as 74% of threat intelligence firms report having difficulty with initial analysis [18].

Studies have demonstrated that putting the human element of cybercrimes into context can greatly enhance attribution and prediction attempts [12]. Although threat actor behavior has been thoroughly studied in the past using anthropological lenses, with an emphasis on cultural drives and values, the psychological and behavioral characteristics of cyber criminals have not been thoroughly studied in a quantitative way. This research, therefore, addresses a significant gap by offering a quantitative analysis of human behavioral traits in cybercrime, contributing to more accurate characterization of criminals.

Moreover, the findings suggest that machine learning models like the one developed in this study could be integrated into existing forensic tools to enhance their analytical capabilities. By automating the detection of anomalous behavior, the model can assist investigators in prioritizing relevant evidence, reducing the time and effort required for manual analysis. This could lead to more efficient investigations and more accurate conclusions, ultimately improving the ability of digital forensics to support law enforcement efforts.

However, while the model demonstrated strong performance, there is room for improvement. Future research could explore the integration of additional data sources, such as network traffic, communication logs, and social media activity, to provide a more holistic view of user behavior. Furthermore, refining the model to reduce computational overhead would make it more accessible for use in resource-limited environments, such as mobile forensics or real-time analysis.

A limitation of this study lies in the heuristic linking of cross-device sessions. While timestamps, cookies, and session tokens were used to correlate activity, no ground truth validation was available. Future work should incorporate labeled cross-device datasets to quantify linkage accuracy and its effect on downstream anomaly detection.

Isolation Forest contamination was set equal to the known anomaly prevalence in training folds. While this yielded strong experimental results, it assumes prior knowledge not available in operational forensic investigations. Future work should explore adaptive or unsupervised contamination estimation methods.

Another promising area for future work is the exploration of cross-platform browser artifact analysis, as users often engage with multiple devices and browsers. The ability to track behavior across different platforms would provide a more complete picture of user activity, further enhancing the model's applicability in digital forensics. Additionally, incorporating more advanced techniques like adversarial learning could help improve the model's resilience to evasion tactics used by sophisticated threat actors who intentionally attempt to conceal their digital footprints.

In conclusion, this research makes a meaningful contribution to the field of digital forensics by demonstrating the effectiveness of machine learning in analyzing browser artifacts for behavior analysis. The model offers a robust, scalable solution for identifying complex patterns of user behavior, providing forensic investigators with a valuable tool for detecting anomalies and uncovering criminal intent. As the digital landscape continues to evolve, so too must the methods used to analyze digital evidence, and this study represents an important step in that direction. With further development and refinement, this approach has the potential to significantly improve the speed, accuracy, and depth of digital forensic investigations, leading to better outcomes for law enforcement and other stakeholders.

To strengthen reproducibility, this work has clarified the evaluation protocol, preprocessing safeguards, and final model configuration. Future research will extend adaptive thresholding, validate cross-device linking, and benchmark against re-implemented baselines on the same datasets to ensure head-to-head comparability. These improvements will enhance both transparency and operational applicability.

## Author contributions

**Conceptualization:** R. K. Gethmi Hasinika Rathnayaka, Amila Nuwan Senarathne.

**Data curation:** D. G. Samesha Navodi Gonawala.

**Formal analysis:** D. G. Samesha Navodi Gonawala, Amila Nuwan Senarathne, S. M. Deemantha Nayanajith Siriwardena.

**Investigation:** R. K. Gethmi Hasinika Rathnayaka, W. Pawani Dananjana, Jithmi Sewwandi Arambawela, S. M. Deemantha Nayanajith Siriwardena.

**Methodology:** R. K. Gethmi Hasinika Rathnayaka, W. Pawani Dananjana, Jithmi Sewwandi Arambawela.

**Project administration:** R. K. Gethmi Hasinika Rathnayaka, Amila Nuwan Senarathne.

**Resources:** D. G. Samesha Navodi Gonawala, W. Pawani Dananjana, Jithmi Sewwandi Arambawela.

**Software:** D. G. Samesha Navodi Gonawala, W. Pawani Dananjana, Jithmi Sewwandi Arambawela.

**Supervision:** Amila Nuwan Senarathne.

**Validation:** S. M. Deemantha Nayanajith Siriwardena.

**Visualization:** W. Pawani Dananjana, Jithmi Sewwandi Arambawela.

**Writing – original draft:** D. G. Samesha Navodi Gonawala, R. K. Gethmi Hasinika Rathnayaka, Amila Nuwan Senarathne, S. M. Deemantha Nayanajith Siriwardena.

**Writing – review & editing:** Amila Nuwan Senarathne, S. M. Deemantha Nayanajith Siriwardena.

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
