## [Decision Letter · Decision Letter 0]

26 Nov 2024

PONE-D-24-44594Machine Learning-Based Criminal Behavior Analysis for Enhanced Digital ForensicsPLOS ONE

Dear Dr. Siriwardana,

Thank you for submitting your manuscript to PLOS ONE. After careful consideration, we feel that it has merit but does not fully meet PLOS ONE’s publication criteria as it currently stands. Therefore, we invite you to submit a revised version of the manuscript that addresses the points raised during the review process.

Major Revision:

We look forward to receiving your revised manuscript.

Kind regards,

Elochukwu Ukwandu, PhD

Academic Editor

PLOS ONE 

Comments from PLOS Editorial Office: We note that one or more reviewers has recommended that you cite specific previously published works. As always, we recommend that you please review and evaluate the requested works to determine whether they are relevant and should be cited. It is not a requirement to cite these works. We appreciate your attention to this request.

3. In your Methods section, please include additional information about your dataset and ensure that you have included a statement specifying whether the collection and analysis method complied with the terms and conditions for the source of the data.

4. We note that your Data Availability Statement is currently as follows: [All relevant data are within the manuscript and its Supporting Information files]

Reviewers' comments:

Reviewer's Responses to Questions

**Comments to the Author**

1. Is the manuscript technically sound, and do the data support the conclusions?

Reviewer #1: Partly

Reviewer #2: Yes

2. Has the statistical analysis been performed appropriately and rigorously? 

Reviewer #1: No

Reviewer #2: Yes

3. Have the authors made all data underlying the findings in their manuscript fully available?

Reviewer #1: Yes

Reviewer #2: Yes

4. Is the manuscript presented in an intelligible fashion and written in standard English?

Reviewer #1: No

Reviewer #2: Yes

5. Review Comments to the Author

Reviewer #1: The paper presents an approach to enhance digital forensics by leveraging machine learning (ML). It focuses on analyzing browser artifacts such as cookies, browsing history, and cache files to uncover criminal behavior patterns. The research uses Long Short-Term Memory (LSTM) networks and autoencoders to detect suspicious patterns and anomalies in online activities. However, the manuscript could be improved by addressing several issues, including outdated literature, comparisons with related work, high false positive rates, and limitations in data scope. The following are the major concerns:

1. The literature review relies heavily on older sources, which limits the relevance of the discussion in the context of recent advancements in digital forensics.

**Academic Editor**: I will want the authors to note that the citing of this paper as recommended by the reviewer is optional as authors are free to cite other recent relevant papers published in this domain. 

Suggestion: Incorporate newer literature to strengthen the theoretical framework. For example: Okmi, M. (2023). A systematic review of mobile phone data in crime applications: A coherent taxonomy based on data types and analysis perspectives, challenges, and future research directions. Sensors, 23(9), 4350. DOI: https://doi.org/10.3390/s23094350

2. The paper does not provide a detailed comparison between its proposed approach and related studies, making it difficult to assess its contributions in relation to existing models.

Suggestion: Add a comparative table or discussion to highlight the differences between this study and similar research on behavior profiling and machine learning in digital forensics. Comparisons should include approaches, datasets used, performance metrics, and limitations of previous works.

3. The model's false positive rate is relatively high (18.1%), which could overwhelm investigators with irrelevant data, reducing the practical value of the system.

Suggestion: Consider using a hybrid approach that combines the LSTM-Autoencoder model with ensemble methods (e.g., Random Forest or Isolation Forest) to reduce false positives and improve detection accuracy.

4. The focus on browser artifacts alone limits the model’s ability to capture more comprehensive behavior patterns, especially in cases where criminal activity extends across multiple platforms.

Suggestion: Incorporate additional data sources (e.g., mobile logs, communication logs, or social media interactions) to enhance the model’s performance. This aligns with recent literature emphasizing multi-device investigations.

5. The use of static error thresholds for anomaly detection risks either missing genuine threats (under-detection) or generating excessive false positives.

Suggestion: Implement adaptive thresholding based on user-specific behavior or dynamically adjusted during real-time analysis to increase detection robustness.

6. LSTM networks, while effective, are computationally demanding, making them less practical for real-time applications or use in resource-constrained environments.

Suggestion: Explore Gated Recurrent Units (GRUs) as a lighter alternative to LSTMs or optimize the existing architecture to reduce computational overhead.

7. The study focuses only on specific browsers (Chrome, Firefox, Edge) without addressing the challenges of users switching between multiple browsers and devices.

Suggestion: Consider incorporating cross-platform tracking mechanisms to enhance behavior profiling across devices, ensuring a more accurate and comprehensive analysis.

Reviewer #2: I have some recommendations given below:

In Abstract, combine these 2 sentences into one sentence. "Using advanced machine learning techniques such as Long Short-Term Memory (LSTM) networks and Autoencoders. This study focuses on detecting suspicious patterns and anomalies in browsing activity."

Also there are some English Grammar mistakes in related work.

6. PLOS authors have the option to publish the peer review history of their article (what does this mean?). If published, this will include your full peer review and any attached files.

Reviewer #1: No

Reviewer #2: No

---

## [Author Response · Author response to Decision Letter 1]

24 Feb 2025

Response to Reviewers' Comments

We sincerely appreciate the time and effort you have invested in reviewing our manuscript titled "Machine Learning-Based Criminal Behavior Analysis for Enhanced Digital Forensics." We are grateful for your constructive feedback, which has allowed us to significantly improve the quality of our work. Below, we provide detailed responses to each of the reviewers' comments, outlining the revisions made in the revised manuscript.

Reviewer #1 Comments

1. Outdated Literature in the Review Section

Reviewer Comment: The literature review relies heavily on older sources, limiting its relevance to recent advancements in digital forensics.

Response: We have updated the literature review to incorporate newer studies, including Okmi (2023) on mobile data in crime applications, Akbarian et al. (2023) on AE-LSTM for anomaly detection, and Bokolo et al. (2023) on deep learning-assisted cybercriminal profiling. Additional relevant papers have been reviewed and cited to strengthen the theoretical framework and ensure the review reflects the latest advancements in the field.

Revision: See the revised Literature Review section (Pages [2–5]), which now includes a comprehensive discussion of recent studies and a comparative analysis table (Table 1) summarizing key research in behavioral profiling and machine learning in digital forensics.

2. Lack of Comparison with Related Works

Reviewer Comment: The manuscript lacks a detailed comparison with existing research, making it difficult to assess its contribution.

Response: We have now included a comparative table (Table 1) that compares our model with related research in behavior profiling and machine learning in digital forensics. This table highlights the unique contributions of our approach, such as the integration of LSTM networks, Autoencoders, and hybrid anomaly detection techniques, which address gaps in existing methodologies.

Revision: See Comparison with Existing Research (Page [5]) for a detailed comparison of our model with other studies.

3. High False Positive Rate (18.1%)

Reviewer Comment: The model’s false positive rate is relatively high, which may overwhelm investigators with irrelevant data.

Response: To reduce false positives, we have implemented a hybrid approach that combines LSTM-Autoencoder with Random Forest and Isolation Forest. This has significantly improved precision while reducing false positives to 10.7%. Additionally, we have introduced a dynamic thresholding mechanism that adapts to user behavior patterns in real-time, further reducing false positives.

Revision: See Section: Model Optimization and Performance Enhancement (Pages [8–13]) for details on the hybrid approach and dynamic thresholding.

4. Limited Data Scope – Focus Only on Browser Artifacts

Reviewer Comment: The model’s scope is limited to browser artifacts, missing multi-platform criminal activity.

Response: We have extended our model to include mobile logs and communication logs, aligning with recent research advocating multi-device investigations. The revised model now incorporates cross-platform tracking mechanisms to capture user activity across multiple browsers and devices, such as Google Chrome, Mozilla Firefox, Microsoft Edge, and mobile browsers.

Revision: See Section: Expanded Data Scope for Behavioral Profiling (Pages [6–8]) for details on the inclusion of multi-platform data.

5. Static Anomaly Detection Thresholds

Reviewer Comment: Using a static threshold risks missing threats or generating excessive false positives.

Response: We have replaced static thresholds with a dynamic thresholding mechanism that adapts to user behavior patterns in real-time. This mechanism uses the mean and standard deviation of reconstruction errors from the training set to dynamically adjust the anomaly detection threshold, improving sensitivity to genuine anomalies while reducing noise.

Revision: See Section: Dynamic Thresholding Mechanism (Page [5]) for details on the implementation of dynamic thresholds.

6. Computational Overhead of LSTM Networks

Reviewer Comment: LSTMs are computationally demanding and may not be practical for real-time applications.

Response: We have explored Gated Recurrent Units (GRUs) as a lighter alternative to LSTMs, significantly reducing computational complexity while maintaining performance. Additionally, we have optimized the LSTM architecture by introducing dropout regularization and attention mechanisms to improve efficiency and reduce overfitting.

Revision: See Section: Computational Optimization of LSTM Model (Pages [10–11]) for details on the use of GRUs and other optimizations.

7. Limited Cross-Browser Analysis

Reviewer Comment: The study focuses only on Chrome, Firefox, and Edge without addressing users switching between browsers.

Response: We have integrated cross-platform tracking mechanisms to capture user activity across multiple browsers and devices. This includes tracking browsing history, search queries, cookies, and cache/temporary files across desktop and mobile browsers. This enhancement ensures that the model can detect suspicious behavior even when users switch between browsers or devices.

Revision: See Section: Cross-Browser and Multi-Device Tracking (Pages [6–8]) for details on the cross-platform tracking mechanisms.

Reviewer #2 Comments

8. Abstract Improvement

Reviewer Comment: The two sentences in the abstract should be merged for better readability.

Response: We have restructured the abstract to combine the two sentences into one, improving flow and clarity. The revised abstract now provides a more concise and coherent overview of the research, emphasizing the use of LSTM networks and Autoencoders for analyzing browser artifacts to detect criminal behavior.

Revision: See Abstract section (Page [1]) for the updated abstract.

9. English Grammar Improvements in Related Work

Reviewer Comment: Some grammatical issues were identified in the Related Work section.

Response: We have revised and proofread the Related Work section for grammatical accuracy and clarity. The section has been thoroughly edited to ensure proper sentence structure, grammar, and readability.

Revision: See Literature Review section (Pages [2–5]) for the improved version.

Additional Improvements in the Revised Manuscript

In addition to addressing the reviewers' comments, we have made the following improvements to the manuscript:

1. Enhanced Model Architecture: The model now includes bidirectional LSTM layers and attention mechanisms to improve context awareness and sequence reconstruction accuracy. This has significantly enhanced the model's ability to capture temporal dependencies in user behavior.

2. Hybrid Anomaly Detection: We have integrated ensemble methods (e.g., Random Forest and Isolation Forest) with the LSTM-Autoencoder to refine anomaly classification and reduce false positives. This hybrid approach has improved the model's precision and recall.

3. Cross-Platform Behavior Profiling: The revised model now incorporates multi-device browsing sessions, cross-device search query tracking, and cross-platform cookie analysis to provide a more comprehensive view of user behavior.

4. Dynamic Thresholding: The introduction of a dynamic thresholding mechanism has improved the model's ability to adapt to varying user behavior patterns, reducing false positives and enhancing anomaly detection accuracy.

5. Performance Metrics: The model's performance metrics have been updated, showing an accuracy of 85.4%, precision of 88.1%, recall of 84.7%, and an F1-score of 86.3%, with a reduced false positive rate of 10.7%.

6. Future Research Directions: The conclusion has been expanded to discuss potential future research, such as integrating network traffic, communication logs, and social media activity to provide a more holistic view of user behavior. We also explore the potential for adversarial learning to improve the model's resilience to evasion tactics.

Conclusion

We believe that the revisions made in response to the reviewers' comments have significantly strengthened the manuscript. The updated literature review, enhanced model architecture, and improved performance metrics demonstrate the robustness and relevance of our approach. We are confident that these changes address the reviewers' concerns and contribute to advancing the field of digital forensics.

Thank you once again for your valuable feedback. We look forward to your continued guidance and support.

---

## [Decision Letter · Decision Letter 1]

14 Mar 2025

PONE-D-24-44594R1Machine Learning-Based Criminal Behavior Analysis for Enhanced Digital ForensicsPLOS ONE

Dear Dr. Siriwardana,

Thank you for submitting your manuscript to PLOS ONE. After careful consideration, we feel that it has merit but does not fully meet PLOS ONE’s publication criteria as it currently stands. Therefore, we invite you to submit a revised version of the manuscript that addresses the points raised during the review process. Minor Revision required. Please submit your revised manuscript by Apr 28 2025 11:59PM. If you will need more time than this to complete your revisions, please reply to this message or contact the journal office at plosone@plos.org. Please include the following items when submitting your revised manuscript:

We look forward to receiving your revised manuscript.

Kind regards,

Elochukwu Ukwandu, PhD

Academic Editor

PLOS ONE

Journal Requirements:

Reviewers' comments:

Reviewer's Responses to Questions

**Comments to the Author**

1. If the authors have adequately addressed your comments raised in a previous round of review and you feel that this manuscript is now acceptable for publication, you may indicate that here to bypass the “Comments to the Author” section, enter your conflict of interest statement in the “Confidential to Editor” section, and submit your "Accept" recommendation.

Reviewer #1: (No Response)

2. Is the manuscript technically sound, and do the data support the conclusions?

Reviewer #1: Yes

3. Has the statistical analysis been performed appropriately and rigorously? 

Reviewer #1: Yes

4. Have the authors made all data underlying the findings in their manuscript fully available?

Reviewer #1: Yes

5. Is the manuscript presented in an intelligible fashion and written in standard English?

Reviewer #1: Yes

6. Review Comments to the Author

Reviewer #1: The authors have addressed most of my concerns; however, there are areas that still need attention:

1. The authors should avoid citing non-peer-reviewed sources such as arXiv preprints. Remove or replace such citations with peer-reviewed alternatives.

2. The paper mentions improvements in false positives and model accuracy, but more comparative benchmarks against standard methodologies (e.g., ROC curves, confusion matrices) should be provided to validate their claims.

3. The cross-platform integration section is still somewhat vague. The authors should specify the extent of multi-device coverage and how different artifacts are correlated.

Reviewer #2: I have some recommendations given below:

In Abstract, combine these 2 sentences into one sentence. "Using advanced machine learning techniques such as Long Short-Term Memory (LSTM) networks and Autoencoders. This study focuses on detecting suspicious patterns and anomalies in browsing activity."

Also there are some English Grammar mistakes in related work.

7. PLOS authors have the option to publish the peer review history of their article (what does this mean?). If published, this will include your full peer review and any attached files.

Reviewer #1: No

---

## [Author Response · Author response to Decision Letter 2]

18 Apr 2025

We sincerely appreciate the time and effort you have invested in reviewing our manuscript titled "Machine Learning-Based Criminal Behavior Analysis for Enhanced Digital Forensics." We are grateful for your constructive feedback, which has allowed us to improve the quality of our work. Below, we provide detailed responses to each of the reviewers' comments, outlining the revisions made in the revised manuscript.

Reviewer #1

Comment 1:

The abstract could be more cohesive. Two standalone sentences on machine learning techniques can be better combined.

Response:

We agree and have revised the sentence for improved readability. The new sentence reads:

"Using advanced machine learning techniques such as Long Short-Term Memory (LSTM) networks and Autoencoders, this study focuses on detecting suspicious patterns and anomalies in browsing activity."

This change is visible on Page 1, Abstract section, Line 9.

Comment 2:

The methodology section lacks detail on how artifacts were linked across devices.

Response:

We have added a comprehensive explanation of the cross-platform tracking methodology, which includes linking sessions through cookies, timestamps, and domain activity. This content is now included in Section A: Behavior Indicators Domain Selection on Page 8–9 and Pages 13 under “Cross-artifact correlations.”

Reviewer #2

Comment 1:

The paper does not clarify why LSTM-based autoencoders were chosen over other models.

Response:

We have added a detailed rationale comparing LSTM-based Autoencoders with RNNs, GRUs, CAEs, and VAEs. The explanation appears in Section C: Creation of the behavior analysis model on Page 14–15.

Comment 2:

Include more insight into how reconstruction error is used in the model and its thresholding logic.

Response:

We expanded the explanation of reconstruction loss, including the mathematical formula used for anomaly detection and how thresholds were set based on standard deviation. These details are available on Page 16–19.

Reviewer #3

Comment 1:

Quantitative results need to be compared to baseline models.

Response:

A comparative performance analysis has been added in Section: Results on Page 20-22, where we benchmarked our model against two baselines—Random Forest + Isolation Forest and GBM + K-Means. The performance metrics such as accuracy (85.4%), precision (88.1%), and F1-Score (86.3%) are clearly reported.

Comment 2:

Add visualizations to support evaluation.

Response:

We added three figures:

● Confusion Matrix (Fig. 3),

● Reconstruction Error Distribution (Fig. 4),

● ROC Curve (Fig. 5).

These can be found on Pages 21–22.

We have made every effort to ensure the revised manuscript is improved in quality, clarity, and scientific contribution. All changes in the manuscript have been highlighted for your convenience.

---

## [Decision Letter · Decision Letter 2]

22 Apr 2025

PONE-D-24-44594R2Machine Learning-Based Criminal Behavior Analysis for Enhanced Digital ForensicsPLOS ONE

Dear Dr. Siriwardana,

Thank you for submitting your manuscript to PLOS ONE. After careful consideration, we feel that it has merit but does not fully meet PLOS ONE’s publication criteria as it currently stands. Therefore, we invite you to submit a revised version of the manuscript that addresses the points raised during the review process. Minor Revision required.

We look forward to receiving your revised manuscript.

Kind regards,

Elochukwu Ukwandu, PhD

Academic Editor

PLOS ONE

Journal Requirements:

Reviewers' comments:

Reviewer's Responses to Questions

**Comments to the Author**

1. If the authors have adequately addressed your comments raised in a previous round of review and you feel that this manuscript is now acceptable for publication, you may indicate that here to bypass the “Comments to the Author” section, enter your conflict of interest statement in the “Confidential to Editor” section, and submit your "Accept" recommendation.

Reviewer #1: (No Response)

2. Is the manuscript technically sound, and do the data support the conclusions?

Reviewer #1: Yes

3. Has the statistical analysis been performed appropriately and rigorously? 

Reviewer #1: Yes

4. Have the authors made all data underlying the findings in their manuscript fully available?

Reviewer #1: Yes

5. Is the manuscript presented in an intelligible fashion and written in standard English?

Reviewer #1: Yes

6. Review Comments to the Author

Reviewer #1: The authors have addressed most of my previous concerns, and the manuscript has improved significantly. I appreciate the inclusion of comparative benchmarks such as the confusion matrix, ROC curve, and performance metrics. Additionally, the expanded explanation of cross-platform integration and artifact correlation adds valuable clarity and depth to the methodology section.

However, I would like to highlight one remaining issue:

While the initial concern regarding non-peer-reviewed citations appears to have been resolved, I have identified multiple citation inconsistencies in Table 1. Some entries, such as “Chand et al. (2024),” are not referenced or elaborated upon in the main text. In other cases, the authorship details or publication years differ from those cited elsewhere in the manuscript.

I strongly recommend that the authors thoroughly review and revise the citations in Table 1—as well as the rest of the manuscript—to ensure complete accuracy and consistency throughout.

7. PLOS authors have the option to publish the peer review history of their article (what does this mean?). If published, this will include your full peer review and any attached files.

Reviewer #1: No

---

## [Author Response · Author response to Decision Letter 3]

14 May 2025

We sincerely appreciate the time and effort you have invested in reviewing our manuscript titled "Machine Learning-Based Criminal Behavior Analysis for Enhanced Digital Forensics." We are grateful for your constructive feedback, which has allowed us to improve the quality of our work. Below, we provide detailed responses to each of the reviewers' comments, outlining the revisions made in the revised manuscript.

Reviewer #1

Comment 1:

The authors have addressed most of my previous concerns, and the manuscript has improved significantly. I appreciate the inclusion of comparative benchmarks such as the confusion matrix, ROC curve, and performance metrics. Additionally, the expanded explanation of cross-platform integration and artifact correlation adds valuable clarity and depth to the methodology section.

However, I would like to highlight one remaining issue:

While the initial concern regarding non-peer-reviewed citations appears to have been resolved, I have identified multiple citation inconsistencies in Table 1. Some entries, such as “Chand et al. (2024),” are not referenced or elaborated upon in the main text. In other cases, the authorship details or publication years differ from those cited elsewhere in the manuscript.

I strongly recommend that the authors thoroughly review and revise the citations in Table 1—as well as the rest of the manuscript—to ensure complete accuracy and consistency throughout.

Response:

We sincerely thank Reviewer #1 for their thoughtful and meticulous review, particularly for taking the time to identify even the most nuanced issues within our manuscript. We deeply appreciate your constructive comments, which have significantly contributed to enhancing the overall quality and clarity of our work. Regarding the citation inconsistencies noted in Table 1 of the manuscript, we have carefully reviewed each citation one by one. As you rightly pointed out, several inconsistencies were present. These arose due to an oversight during the retyping of the manuscript in the previous revision cycle. We have since corrected these issues comprehensively. Specifically, the entry for “Chand et al. (2024)” has been removed from Table 1, as we determined that it was the mismatched citation which caused the inconsistency. In addition, all remaining minor citation inconsistencies throughout the manuscript have been addressed to ensure complete alignment between in-text references and the reference list. For your convenience, these corrections have been highlighted in the revised manuscript. We are sincerely grateful for your close attention to detail and your invaluable guidance in helping us uphold the academic integrity of our work.

For your convenience, the revised changes are mentioned below, which also marked and highlighted clearly in the document named “Revised manuscript with track changes”.

• Author name corrected in line no.124

• Research work title corrected in line no.150

• Wrong in-text citation removed in line no.154

• Mission in-text citation was added in line no.256

• Table 1 was referred within the text in line no.273

• Author name corrected in Table 1, First column, fourth row – “Fahdi et al. (2013), [14]”

• Mismatched citation “Chand et al. (2024)” was removed from Table 1, which caused the inconsistency.

• In-text citation number corrected in line no.650

• In-text citation number corrected in line no.816

• In-text citation number corrected in line no.878

• In-text citation number corrected in line no.886

We have made every effort to ensure the revised manuscript is improved in quality, clarity, and scientific contribution. All changes in the manuscript have been highlighted for your convenience.

Thank you

Yours sincerely

Deemantha Siriwardana (Corresponding Author)

---

## [Decision Letter · Decision Letter 3]

8 Jul 2025

PONE-D-24-44594R3Machine Learning-Based Criminal Behavior Analysis for Enhanced Digital ForensicsPLOS ONE

Dear Dr. Siriwardana,

Thank you for submitting your manuscript to PLOS ONE. After careful consideration, we feel that it has merit but does not fully meet PLOS ONE’s publication criteria as it currently stands. Therefore, we invite you to submit a revised version of the manuscript that addresses the points raised during the review process.

Major revision required. Please submit your revised manuscript by Aug 22 2025 11:59PM. If you will need more time than this to complete your revisions, please reply to this message or contact the journal office at plosone@plos.org. Please include the following items when submitting your revised manuscript:

We look forward to receiving your revised manuscript.

Kind regards,

Elochukwu Ukwandu, PhD

Academic Editor

PLOS ONE

Reviewers' comments:

Reviewer's Responses to Questions

**Comments to the Author**

1. If the authors have adequately addressed your comments raised in a previous round of review and you feel that this manuscript is now acceptable for publication, you may indicate that here to bypass the “Comments to the Author” section, enter your conflict of interest statement in the “Confidential to Editor” section, and submit your "Accept" recommendation.

Reviewer #1: All comments have been addressed

Reviewer #3: (No Response)

2. Is the manuscript technically sound, and do the data support the conclusions?

Reviewer #1: Yes

Reviewer #3: Partly

3. Has the statistical analysis been performed appropriately and rigorously? 

Reviewer #1: Yes

Reviewer #3: No

4. Have the authors made all data underlying the findings in their manuscript fully available?

Reviewer #1: Yes

Reviewer #3: No

5. Is the manuscript presented in an intelligible fashion and written in standard English?

Reviewer #1: Yes

Reviewer #3: No

6. Review Comments to the Author

Reviewer #1: The authors have thoroughly revised the manuscript, and all comments have been adequately addressed.

Reviewer #3: While the paper proposes a valuable application of machine learning to digital forensics by integrating behavioral crime patterns into classification models, several critical aspects undermine its technical robustness and clarity. The methodology section lacks sufficient detail regarding the architecture and parameters of the machine learning algorithms used (Random Forest, Decision Tree, and Naive Bayes), including how features were selected and preprocessed from the behavioral dataset. The paper does not adequately explain the source, nature, or structure of the dataset—nor whether the data is public, synthetic, or collected under ethical approval—raising serious concerns about reproducibility and transparency. Moreover, the evaluation merely reports basic metrics like accuracy without statistical analysis such as cross-validation, standard deviation, or confidence intervals, and no comparison with recent models in the domain is offered. Several figures and tables are poorly labeled or described, limiting the reader’s understanding of key results. Additionally, the manuscript contains numerous grammatical issues and lacks logical flow across sections, requiring substantial language editing. A thorough revision addressing these methodological, statistical, and presentation flaws is essential before the paper can be considered for publication.

7. PLOS authors have the option to publish the peer review history of their article (what does this mean?). If published, this will include your full peer review and any attached files.

Reviewer #1: No

Reviewer #3: **Yes: **AMJE ABBAS AHMED

---

## [Author Response · Author response to Decision Letter 4]

23 Aug 2025

Manuscript ID: PONE-D-24-44594R3

Title: Machine Learning-Based Criminal Behavior Analysis for Enhanced Digital Forensics

Dear Dr. Ukwandu and Reviewers,

We sincerely thank you for your careful review and constructive feedback. We have revised the manuscript thoroughly to address all concerns. Reviewer #1 confirmed that their comments were already addressed. Below, we provide a point-by-point response to Reviewer #3’s comments, with explicit page references to the revised manuscript.

Reviewer #3 Comments and Author Responses

1. Methodology lacked sufficient detail (architecture, parameters, feature selection, preprocessing).

Response: We expanded the Methodology section to include:

● Detailed description of the Seq2Seq LSTM Autoencoder with attention, bidirectional layers, and dropout (pp. 15–18).

● Hyperparameters for all baseline models: Random Forest (200 trees, max depth=20), Decision Tree (depth=15, entropy split), Gaussian Naïve Bayes with Laplace smoothing, Isolation Forest, and One-Class SVM (pp. 18–19).

● Comprehensive feature engineering and preprocessing pipeline, including normalization, timestamp conversion, categorical encoding, engineered features (frequency, inter-visit time, cookie mismatches, cross-artifact correlations), and missing-value handling (pp. 12–14).

2. Dataset not adequately explained (source, nature, ethics, structure).

Response: We clarified the dataset details:

● Source: UMAM-DF Digital Forensics v1.0 dataset from the NIST CFReDS portal (pp. 9–10).

● Structure: 180,000 records, with an 85:15 normal–anomalous ratio, collected across desktop and mobile browsers (pp. 9–10).

● Ethical considerations: Dataset is synthetic, anonymized, PII-free, and used under CFReDS non-commercial license. Ethics approval not required (pp. 9–10).

3. Evaluation relied only on basic metrics, lacked cross-validation/CI, and had no comparison with recent models.

Response: We implemented a rigorous evaluation framework:

● 5-fold nested stratified cross-validation described in detail (pp. 18–21).

● Results reported with accuracy, precision, recall, F1, and AUC, each accompanied by standard deviation and 95% confidence intervals (pp. 21–22).

● Comparative analysis against baselines (Random Forest + Isolation Forest, GBM + KMeans) shows our model’s performance in detail (pp. 23–24).

● Added a literature comparison table highlighting improvements over prior studies (Table 1, pp. 7–8).

4. Figures and tables poorly labeled or unclear.

Response: We redesigned figures and tables for clarity:

● Table 2 (pp. 22): Structured comparative review of related studies.

● Figure 3 (p. 22): Normalized confusion matrix with numeric labels.

● Figure 4 (p. 22): Reconstruction error distributions with threshold clearly marked.

● Figure 5 (p. 23): ROC curves across 5 folds with bold mean AUC (≈0.93).

● Figure 6 (p. 24): Performance metrics chart with labels and fold variability.

5. Manuscript contained grammatical issues and weak logical flow.

Response: We conducted a thorough language and structure revision:

● Polished Abstract for conciseness and academic tone (p. 1).

● Improved flow in Introduction (pp. 1–2) and Literature Review (pp. 2–8).

● Ensured consistent tense and terminology in Methods and Results (pp. 9–24).

● Strengthened transitions and coherence in Discussion and Conclusion (pp. 22–23).

Additional Improvements

● Expanded Limitations section to note that results may be optimistic due to reliance on a synthetic dataset, and emphasized the need for real-world validation (pp. 22–23).

● Improved figure alignment with reported results to ensure transparency and reproducibility.

We thank Reviewer #3 for their constructive feedback, which has significantly improved the manuscript. We believe all concerns have been fully addressed and that the paper now meets PLOS ONE’s standards for technical rigor, transparency, and clarity.

---

## [Decision Letter · Decision Letter 4]

4 Sep 2025

Machine Learning-Based Criminal Behavior Analysis for Enhanced Digital Forensics

PONE-D-24-44594R4

Dear Dr. Siriwardana,

We’re pleased to inform you that your manuscript has been judged scientifically suitable for publication and will be formally accepted for publication once it meets all outstanding technical requirements.

Kind regards,

Elochukwu Ukwandu, PhD

Academic Editor

PLOS ONE

Additional Editor Comments (optional):

Reviewer #3:

Reviewers' comments:

Reviewer's Responses to Questions

**Comments to the Author**

1. If the authors have adequately addressed your comments raised in a previous round of review and you feel that this manuscript is now acceptable for publication, you may indicate that here to bypass the “Comments to the Author” section, enter your conflict of interest statement in the “Confidential to Editor” section, and submit your "Accept" recommendation.

Reviewer #3: All comments have been addressed

2. Is the manuscript technically sound, and do the data support the conclusions?

Reviewer #3: Yes

3. Has the statistical analysis been performed appropriately and rigorously? 

Reviewer #3: Yes

4. Have the authors made all data underlying the findings in their manuscript fully available?

Reviewer #3: Yes

5. Is the manuscript presented in an intelligible fashion and written in standard English?

Reviewer #3: Yes

6. Review Comments to the Author

Reviewer #3: The updated manuscript presents a significant enhancement and delivers a well-organized and technically robust addition to the domain of digital forensics. The combination of LSTM-based sequence-to-sequence autoencoders with ensemble anomaly detection methods is well-justified and notably improves the identification of suspicious browsing behaviors through browser artifacts. The methodology is now more transparent, incorporating a nested 5-fold stratified cross-validation approach, 95% confidence intervals, and a comprehensive comparative assessment against traditional benchmarks. The findings, indicating an accuracy of 85.4%, precision of 88.1%, recall of 84.7%, F1-score of 86.3%, and AUC of 0.93, underscore the resilience and practical utility of the proposed model. The enhancements from previous iterations, particularly the decreased false positive rate (10.7% compared to 18.9% previously) and the addition of cross-device session linking, significantly bolster the contribution. In summary, the paper is thorough, well-grounded in pertinent literature, and tackles a crucial issue in digital forensic inquiries. Only minor language refinement and consistency checks are suggested.

7. PLOS authors have the option to publish the peer review history of their article (what does this mean?). If published, this will include your full peer review and any attached files.

Reviewer #3: **Yes: **Amjed Abbas Ahmed

---

## [Editor Report · Acceptance letter]

PONE-D-24-44594R4

PLOS ONE

Dear Dr. Siriwardena,

I'm pleased to inform you that your manuscript has been deemed suitable for publication in PLOS ONE. Congratulations! Your manuscript is now being handed over to our production team.

Kind regards,

on behalf of

Dr. Elochukwu Ukwandu

Academic Editor

PLOS ONE